# Effect of miR-34a on the expression of clock and clock-controlled genes in DLD1 and Lovo human cancer cells with different backgrounds with respect to p53 functionality and 17β-estradiol-mediated regulation

**Roman Moravčík, Soňa Olejárová, Jana Zlacká, Iveta Herichová** *

Faculty of Natural Sciences, Department of Animal Physiology and Ethology, Comenius University, Bratislava, Slovak Republic

* herichova1@uniba.sk

**Data Availability Statement:** The dataset used in the manuscript is stored in the data repository

## Abstract

The small non-coding RNA miR-34a is a p53-regulated miRNA that acts as a tumour suppressor of colorectal cancer (CRC). Oncogenesis is also negatively influenced by deregulation of the circadian system in many types of tumours with various genetic backgrounds. As the clock gene *per2* was recently recognized as one of the target genes of miR-34a, we focused on the miR-34a-mediated influence on the circadian oscillator in CRC cell lines DLD1 and LoVo, which differ in their p53 status. Previously, a sex-dependent association between the expression of *per2* and that of miR-34a was demonstrated in CRC patients. Therefore, we also investigated the effect of 17β-estradiol (E2) on miR-34a oncostatic functions. miR-34a mimic caused a pronounced inhibition of *per2* expression in both cell lines. Moreover, miR-34a mimic significantly inhibited *bmal1* expression in LoVo and *rev-erbα* expression in DLD1 cells and induced *clock* gene expression in both cell lines. miR-34a mimic caused a pronounced decrease in *sirt1* and *cyclin D1* expression, which may be related to the inhibition of proliferation observed after mir-34a administration in DLD1 cells. E2 administration inhibited the migration and proliferation of DLD1 cells. E2 and miR-34a, when administered simultaneously, did not potentiate each other's effects. To conclude, miR-34a strongly influences the expression of components of the circadian oscillator without respect to p53 status and exerts its oncostatic effects via inhibition of *sirt1* and *cyclin D1* mRNA expression. E2 administration inhibits the growth of DLD1 cells; however, this effect seems to be independent of miR-34a-mediated action. With respect to the possible use of miR-34a in cancer treatment, clock genes can be considered as off-target genes, as changes in their expression induced by miR-34a treatment do not contribute to the oncostatic functions of miR-34a. Possible ambiguous oncogenic characteristics should be taken into consideration in future clinical studies focused on miR-34a.

Zenodo and has a digital object identifier 10.5281/zenodo.8238994.

**Funding:** IH The Slovak Research and Development Agency https://www.apvv.sk/?lang=en APVV-16-0209 APVV-20-0241 Scientific Grant Agency of the Ministry of Education, Science, Research and Sport of the Slovak Republic https://www.minedu.sk/vedecka-grantova-agentura-msvvas-sr-a-sav-vega/ VEGA 1/0455/23 The funders had no role in study design, data collection and analysis, decision to publish, or preparation of the manuscript.

**Competing interests:** The authors have declared that no competing interests exist.

## Introduction

Colorectal cancer (CRC) is one of the leading causes of cancer death in Europe and the U.S.A. [1,2]. It is the third most frequently diagnosed oncological disease and has the second highest mortality rate [3]. Moreover, an increase in CRC incidence in patients below 50 years of age recently emerged [4,5]. Therefore, novel strategies for CRC treatment are being studied extensively. Among them, gene silencing and personalisation of therapy are promising approaches to replace or modify classical disease management.

Interestingly, a higher incidence of colorectal cancer has been demonstrated in male than in female patients [6,7]. Sex-dependent features of CRC progression are attributed, in addition to factors such as inflammatory diseases occurrence, genetic predisposition and/or social habits, as well as differences in levels of sex hormones between males and females [8–10]. In particular, 17β-estradiol (E2) has been shown to exert beneficial effects with respect to CRC [11–13].

E2 signalling is mediated via two nuclear receptors, ESR1 (ERα) and ESR2 (ERβ), and one membrane-bound G protein-coupled receptor-1 (GPER1). The most of beneficial E2 effects in the gastrointestinal tract of CRC patients are executed by ESR2 receptors [14]. The first implications of E2's capacity to influence CRC management were based on epidemiological studies including patients subjected to hormone replacement therapy where E2 administration showed a negative association with CRC incidence [15,16]. A protective effect of E2 was demonstrated in ovariectomised (OVX) mice bearing a mutation in the Apc gene that produced many more intestinal tumours compared to those that were not subjected to OVX. In this model, E2 administration reversed the effect of OVX and induced ESR2 expression [17]. Similarly, APC-deficient mice with deletion of ESR2 receptors produced larger adenomas compared to those with wild-type ESR2 [18]. The protective role of E2 mediated by ESR2 receptors was also demonstrated in azoxymethane-treated OVX mice [19]. Implantation of SW480 cells overexpressing ESR2 to immunodeficient mice issued in lower tumour growth compared to mice implanted with control SW480 cells [20]. In male and female mice, E2 significantly reduced the number of azoxymethane/dextran sulphate sodium induced tumours and attenuated inflammation [21].

The oncostatic potential of E2 has been shown in a wide range of cellular models of colorectal cancer (for a review see [22]). Protective E2 effects mediated by ESR2 receptors are usually executed by induction of DNA repair, initiation of cell cycle arrest, apoptosis and/or epigenetic control mediated via miRNA and DNA methylation [13].

As cell cycle regulation is a complex process, E2 interferes with other control systems in this respect. In particular, the circadian system seems to influence cancer cell progression and patient survival in a sex-dependent manner [12,23,24].

The generation of circadian oscillators is based on a feedback loop of mRNA transcription and translation of protein products of three homologous *per* (*per1*, *per2* and *per3*) and two homologous *cry* (*cry1* and *cry2*) clock genes. Clock gene expression is induced by a heterodimer composed of the transcription factors BMAL1 and CLOCK via E-box. After accumulation of PER and CRY proteins in the cytoplasm, they create an inhibitory complex that, after translocation to the nucleus, interferes with the stimulatory action of the heterodimer CLOCK/BMAL1 and suppresses their corresponding mRNA transcription [25]. Clock genes can influence the transcriptome directly via E-box [26] or indirectly via clock-controlled genes, including *rev-erbα* [27], *sirt1* [28] and *cyclin D1* (*ccnd1*) [29–31].

Numerous studies have reported changes in clock gene expression in CRC patients [32–35]. Moreover, it was shown that CRC-related dysregulation of the circadian molecular feedback loop exerts sex-dependent effects. Expression of the clock genes *cry2* and *per2* showed a more pronounced decrease in tumours compared to adjacent healthy tissue in males than in

females [12]. Expression of *cry1* differed between left- and right-sided CRC tumours, and these differences exerted sex-dependent patterns [36]. Higher *cry1* expression was observed in colorectal liver metastases [37] and in CRC tumour tissue of males compared to that of females [33,34]. In female CRC patients, better five-year survival was associated with low *cry1* and *cry2* expression, and this association was not observed in males [12,36]. The tumour oncogenic role is usually attributed to the *cry1* gene [34,38,39], whereas *per2* is recognized as a tumour suppressor [40–43].

Regulatory interference between the circadian system and E2 signalling has been described previously. E2 effects on the circadian oscillator are mediated via the ERE region that was identified in *per1* [44], *per2* [44–46] and *clock* [47], and an ESR1 binding site was revealed near the *bmal1* gene promoter [48]. E2 signalling also influences the 24-h pattern of clock gene expression; E2 administration caused a phase advance in *per2* expression in the rat colon. In the same tissue, E2 induced an increase in *bmal1* expression in the middle of the D phase of the LD cycle. This effect was confirmed under *in vitro* conditions, where E2 administration induced *bmal1* mRNA expression in DLD1 cells [49].

In addition to E-box and E2-mediated control, clock gene expression is under the regulatory influence of microRNAs (miRNAs). miRNAs are small non-coding RNAs that usually inhibit gene expression after targeting mRNA [50–52].

miR-34a is an extensively studied miRNA that was identified as a tumour suppressor [53,54] and whose expression is induced by p53 [55]. The expression of *per2*, a key component of the molecular feedback loop [25], is also modulated by p53 [56]. As the majority of CRC tumours express mutated forms of p53 [57], the aim of the present study was to reveal whether p53 status influences the effects of miR-34a on clock and clock-controlled gene expression involved in cell cycle control.

To achieve the above-mentioned aim, we analysed for the first time the effects of miR-34a on the circadian oscillator using the cell lines DLD1 and LoVo CRC, which differ with respect to p53 functionality [58]. Our experimental approach was based on manipulation of intracellular miR-34a levels and consequent changes in clock and clock-controlled gene expression, cell proliferation and the rate of wound healing. The effect of E2 on DLD1 cells was investigated to reveal possible interference between miR-34a and E2-mediated regulation.

miR-34a caused p53-independent inhibition of *per2* expression and changes in the expression of other clock genes (*bmal1*, *clock*). Ectopic miR-34a also inhibited the expression of the clock-controlled genes *rev-erbα*, *sirt1* and *cyclin D1*. Our results do not support a synergic effect of miR-34a and E2 in regulation of DLD1 cell progression.

## Materials and methods

### Cell lines

Human colorectal carcinoma cell lines DLD1 and LoVo were obtained from the ATCC (USA). DLD1 cells were cultured in RPMI 1640 GlutaMax medium (Gibco, USA) supplemented with 10% FBS, 100 U/mL penicillin and 100 μg/mL streptomycin (Biosera, France). LoVo cells were cultured in F-12K medium (Bioconcept, Switzerland) supplemented with 10% FBS, 100 U/mL penicillin and 100 μg/mL streptomycin (Biosera, France). Cells were cultivated at 37°C in a HF90 humidified incubator (Heal Force, China) containing 5% $CO_2$. Experiments were performed three times.

### miRNA used in the study and their transfection

Transfection of miRNA was performed using mirVana™ miRNA mimic hsa-miR-34a (Invitrogen, ThermoFisher Scientific, USA) or mirVana™ miRNA mimic Negative control (Invitrogen,

ThermoFisher Scientific, USA). For gene expression analysis, cells were seeded on 12-well plates (DLD1 $10^6$ cells /well; LoVo 5 x $10^5$ cells/well), and immediately after seeding, the transfection mixture consisting of Lipofectamine RNAiMAX transfection reagent (8 μl/ml; Invitrogen, ThermoFisher Scientific, USA), Opti-MEM$^{TM}$(Invitrogen, ThermoFisher Scientific, USA) and micro-RNA mimic or their negative control at a concentration of 50 nM were added to the cells and incubated for 48 h.

## RNA isolation and PCR

Total RNA isolation from DLD1 or LoVo cells was performed using RNAzol RT (Molecular Research Center, USA). In particular, we used the 'Protocol for isolation of large RNA and small RNA fraction' that allows the separation of long and short RNA molecules. The RNA concentration and purity were determined using a Simply Nano spectrometer (Ge Healthcare, USA). To synthesise cDNA from mRNA, we used 0.3 μg of the fraction containing long molecules and a mixture consisting of ImProm-II reverse transcriptase, ImProm reaction buffer, 10 mM dNTP, RNasin ribonuclease inhibitor and random hexamer primers (Promega, USA). Quantitative PCR was performed with a QuantiTec SYBR Green kit (Qiagen, Germany) and the CFX Connect Real-Time Detection System (Bio-Rad Laboratories, USA). Data were analysed using CFX Manager Software (Bio-Rad Laboratories, USA) and normalized to *u6*, *rplp13a* and/or *β-actin*. The list of primers is provided in S1 Table.

miRNAs from the DLD1 cell line were isolated using RNAzol RT (Molecular Research Center). The miRNA concentration and purity were determined using a Simply Nano spectrometer (Ge Healthcare, USA). Synthesis of cDNA was performed after polyadenylation (Poly(A) Tailing Kit, ThermoFisher Scientific, USA) of isolated miRNAs using a reaction mixture consisting of ImProm-II reverse transcriptase, ImProm reaction buffer, 10 mM dNTP, RNasin Ribonuclease Inhibitor (Promega, USA) and a primer with a universal tag [59]. Quantitative PCR was performed using QuantiTec SYBR Green (Qiagen, Germany) in the CFX Connect Real-Time Detection System (Bio-Rad Laboratories, USA). Data were analysed using CFX Manager Software (Bio-Rad Laboratories, USA) [60]. The sequences of primers for the measurement of miR-34a-5p are provided in S1 Table.

## Fluorescent staining of PER2

DLD1 cells were transfected using mirVana™ hsa pre-miR-34a-5p (Invitrogen; ThermoFisher Scientific, USA) or mirVana™ pre-miRNA mimic Negative control (Invitrogen; ThermoFisher Scientific, USA). The transfection mixture consisted of Lipofectamine RNAiMAX Reagent (8 μl/ml; Invitrogen; ThermoFisher Scientific, USA), Opti-MEMTM (Invitrogen; ThermoFisher Scientific, USA) and pre-micro-RNA or pre-negative control (Invitrogen; ThermoFisher Scientific, USA, 100 nM). Cells were seeded into an eight-well chamber slide that was covered with 0.1% gelatine. After 24 h of incubation, fixation was performed with 4% paraformaldehyde (PFA). After 2 min of incubation, the 4% PFA was discarded and replaced with 300 μl 2% PFA and incubated for 20 min at laboratory temperature. After fixation, cell monolayers were washed with Dulbecco′s phosphate buffered saline (DPBS). Permeabilization of DLD1 cells was performed by adding 0.1% Triton, and the chamber slide was gently mixed for 2 min. This step was repeated four times. After permeabilization, the slide was washed three times with DPBS. Background staining was blocked by incubation of cells with 10% bovine serum albumin for 30 min at laboratory temperature. Next, monolayers were washed with DPBS three times for 5 min. The PER2 primary antibody (sc-377290, Santa Cruz Biotechnology, US) was added to each well in a final concentration of 100 nM, and the cells were incubated overnight at 4˚C. Then, DLD1 cells were washed with DPBS for 5 min. The m-IgGk

secondary antibody (sc-516176, Santa Cruz Biotechnology, US) was added to each well at a concentration of 100 nM, and the cells were incubated for 60 min at laboratory temperature in the dark. After incubation, the cells were washed with DPBS three times. Finally, a drop of mounting medium with DAPI (ibidi, Germany) was added to each well, and the cells were visualised using an inverse fluorescent microscope NIB-100F (NOVEL, China) and pro-gramme BEL Capture 3.2 (NOVEL, China). The intensity of fluorescence was measured using ImageJ 1.53a (NIH, USA).

## Evaluation of the metabolic activity and viability of cells by MTS test

**Effect of E2.** DLD1 cells were seeded in 96-well plates at a density of 2 x $10^4$ cells/well. The next day, the cells were treated with E2 at concentrations of 10, 20, 40, 60, 80 or 100 nM for 24 h. After treatment, the cells were incubated with 10 μM of MTS reagent according to the manufacturer's instructions (CellTiter 96$^{®}$ Aqueous MTS Reagent, Promega, USA). The absorbance was measured at a wavelength of 490 nm (Epoch2, BioTek instruments, USA).

**Combined effect of E2 and miR-34a.** DLD1 cells were seeded in a 96-well plate at a den-sity of 2 x $10^4$ cells/well. Immediately after seeding, the cells were transfected with mirVana™ miRNA mimic has-miR-34a-5p (Invitrogen; ThermoFisher Scientific, USA) or mirVana™ miRNA mimic Negative control (Invitrogen; ThermoFisher Scientific, USA), according to the manufacturer's instructions. The transfection mixture consisted of Lipofectamine RNAiMAX transfection reagent (8 μl/ml; Invitrogen; ThermoFisher Scientific, USA), Opti-MEM$^{TM}$ (Invi-trogen; ThermoFisher Scientific, USA), and micro-RNA or negative control at a concentration of 50 nM was added to the cells and incubated for 48 h. After the transfection, E2 at a concen-tration of 100 nM was added to the medium for 24 h. Next, the cells were incubated with 10 μM of MTS reagent according to the manufacturer's instructions (CellTiter 96$^{®}$ Aqueous MTS Reagent, Promega, USA). The absorbance was measured at a wavelength of 490 nm (Epoch2, BioTek instruments, USA).

## Wound healing assay

A wound healing assay was used to determine migration of DLD1 cells after miR-34a transfec-tion or after E2 treatment. The cells were cultured in 24-well plates. After 24 h of incubation with miR-34a or after incubation with E2 (1 nM, 10 nM, 100 nM) for 24 h or 48 h, the mono-layer of each well was wounded with a 200 μl pipette tip. The recovered area was observed with a NIB-100F inverted microscope (NOVEL, China) and recorded at time zero and after 24 h and 48 h of incubation. Photographs were taken from the same areas as those recorded at time zero and evaluated by ImageJ 1.53a (NIH, USA).

## Statistical analysis

All results are presented as the mean ± SEM in arbitrary units (a.u.). Statistical differences among groups of independent samples were determined by Student's unpaired t-test or by one-way ANOVA followed by Tukey's *post hoc* test or Dunnett's multiple comparison test. A *p*-value < 0.05 was considered statistically significant. ANOVA and t-tests were calculated using Graph Pad Prism 6 (GraphPad Software, Inc., San Diego, California, USA). Image analy-sis was carried out using ImageJ software [61].

## Results

To determine the success of miRNA mimic transfection, the expression of miR-34a in control and miR-34a-transfected cells was measured. The expression of miR-34a was significantly

higher in samples transfected with the miR-34a mimic ($p < 0.001$) compared to the control (S1 Fig).

After validation of transfection, the expression of the clock genes *per2*, *clock* and *bmal1* were analysed. In response to miR-34a administration, the expression of the *per2* gene was significantly decreased in DLD1 ($p < 0.05$) (Fig 1A) and LoVo cell lines ($p < 0.001$) (Fig 1B).

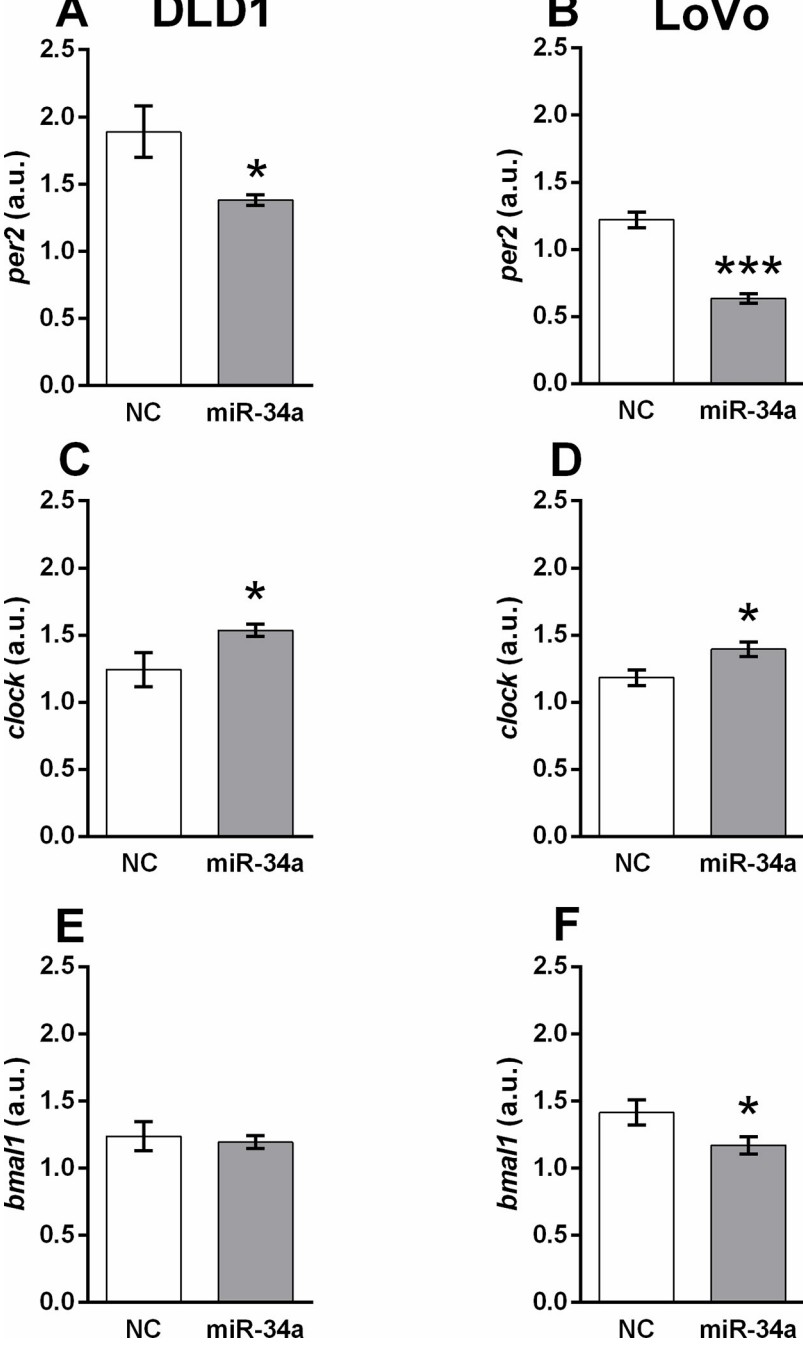

**Fig 1.** Expression of clock genes in DLD1 (A, C, E) and LoVo (B, D, F) cell lines. mRNA levels of *per2* (A, B), *clock* (C, D) and *bmal1* (E, F) were analysed 48 h after miR-34a mimic transfection. Results are displayed as mean ± SEM (n = 6). Statistical differences were determined by unpaired Student's t-test (*$p < 0.05$, ***$p < 0.001$). NC–negative control.

Fluorescent staining of *per2* protein in DLD1 cells (S2A and S2B Fig) supports these findings. The intensity of fluorescence was significantly decreased in pre-miR-34a-transfected cells (S2C Fig).

Unlike *per2*, we detected a significant increase in *clock* gene expression after miR-34a transfection in DLD1 and LoVo cells ($p < 0.05$) (Fig 1C and 1D, respectively).

The expression of *bmal1* was not significantly altered in response to miR-34a treatment in DLD1 cells (Fig 1E); however, a significant decrease in *bmal1* expression was revealed in LoVo cells after miR-34a administration compared to the control ($p < 0.05$) (Fig 1F).

The expression of *sirt1* significantly decreased after miR-34a mimic transfection in DLD1 cells (Fig 2A), ($p < 0.01$) and LoVo cells (Fig 2B) ($p < 0.05$).

Expression of the clock-controlled gene *rev-erbα* was significantly decreased after miR-34a mimic transfection in DLD1 cells (Fig 2C) ($p < 0.05$). In the LoVo cell line, we observed only a slight decreasing trend after miR-34a administration (Fig 2D).

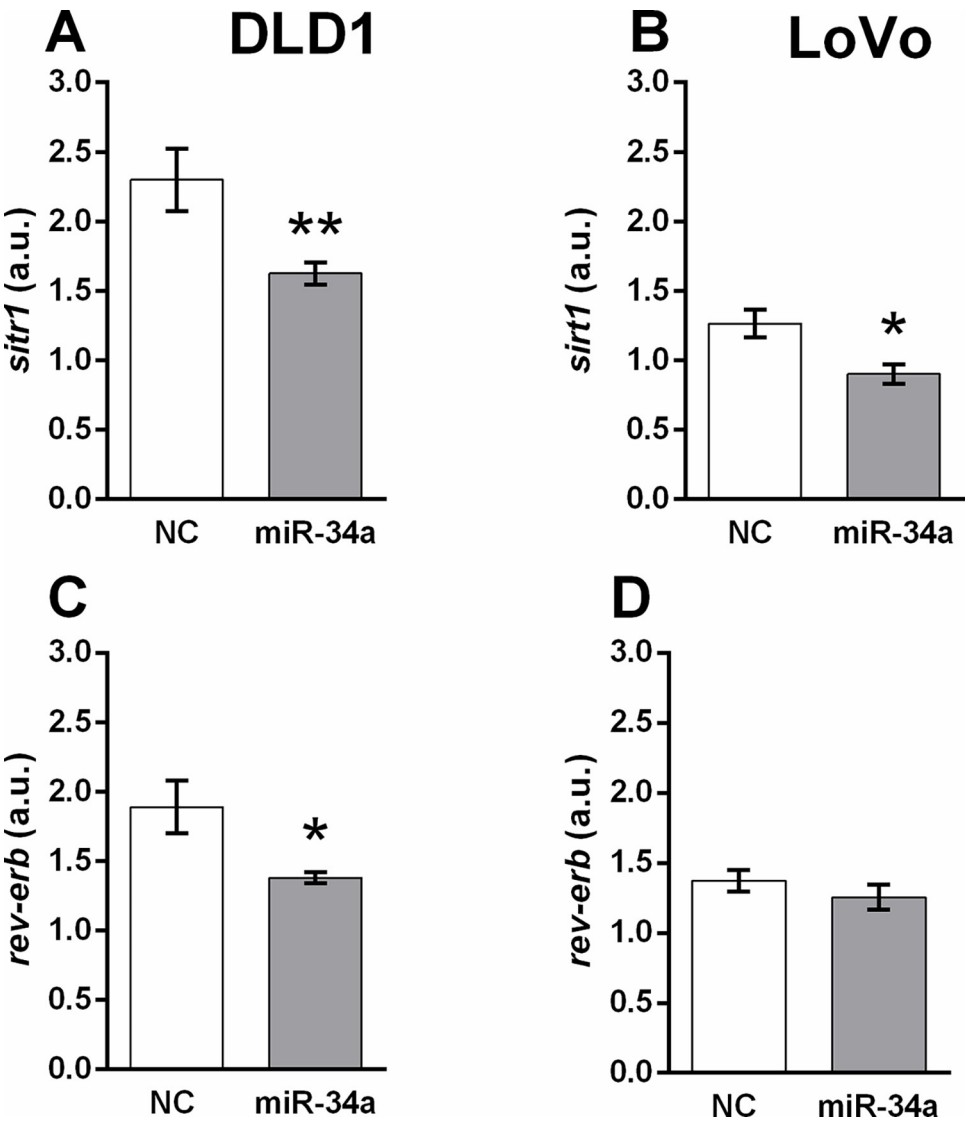

**Fig 2.** Expression of the clock-controlled genes *sirt1* (A, B) and *rev-erbα* (C, D) 48 h after transfection of DLD1 (A, C) and LoVo (B, D) cells with the miR-34a mimic or the corresponding control. Results are displayed as mean ± SEM (n = 6). Statistical differences were determined by unpaired Student's t-test (*$p < 0.05$, **$p < 0.01$). NC–negative control.

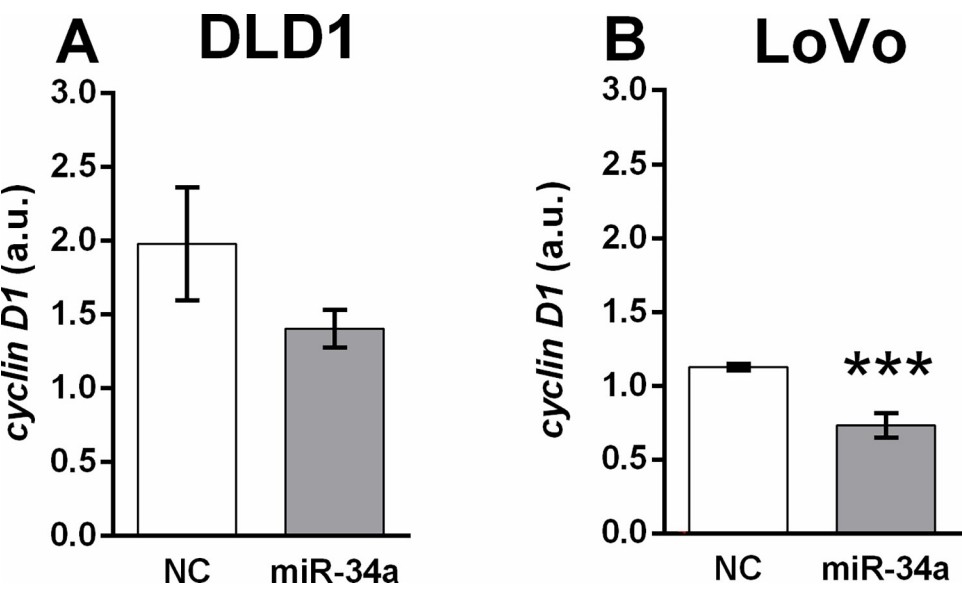

**Fig 3.** Expression of *cyclin D1* 48 h after transfection of DLD1 (A) and LoVo (B) cell lines with the miR-34a mimic or the corresponding control. Results are displayed as mean ± SEM (n = 6). Statistical differences were determined by unpaired Student's t-test (*$p < 0.05$). NC–negative control.

Transfection of the miR-34a mimic-induced decreasing trend in *cyclin D1* expression in DLD1 ($p = 0.1559$) (Fig 3A). A significant decrease in *cyclin D1* expression was observed in the LoVo cell line in response to miR-34a treatment (***$p < 0.001$) (Fig 3B).

Two-day incubation of DLD1 cells with E2 administered in concentrations from 0.1 nM to 100 nM caused a significant increase in *cyclin D1* mRNA expression in DLD1 cells exposed to 100 nM E2 compared to control cells (Fig 4).

The effect of ectopic miR-34a on migration of DLD1 cells was analysed by a wound closure assay. The migration of transfected cells was recorded 24 h after transfection. We observed a trend ($p = 0.115$) of decreasing migration after miR-34a administration in DLD1 cells (Fig 5).

To determine possible reciprocal regulatory interference between miR-34a and E2, we evaluated the expression of *esr2* mRNA in control and miR-34a-treated DLD1 cells. According to our results, miR-34a mimic transfection did not affect *esr2* expression (Fig 6A).

We also tested the possible effect of E2 on the levels of miR-34a in DLD1 cells. E2 administered in four concentrations up to 100 nM did not significantly influence the levels of mature miR-34a after 48 h of incubation with E2 (Fig 6B).

Subsequently, we determined the effect of E2 on DLD1 cell proliferation and migration. We found that E2 administration in concentrations of 60, 80 and 100 nM significantly decreased the proliferation activity of DLD1 cells compared to control cells (Fig 7).

To further elucidate possible regulatory interference between miR-34a and E2, the anti-proliferative effect of E2 was tested with or without the presence of ectopic miR-34a. E2 at a concentration of 100 nM was added to the cell culture 48 h after miR-34a transfection. The absorbance at 490 nm was measured when the cells were exposed to E2 for 24 h. Transfection of the mir-34a mimic significantly decreased the proliferation of DLD1 cells (Fig 8). However, the simultaneous administration of miR-34a and E2 did not change the proliferation rate compared to cells treated with only miR-34a (Fig 8).

The effect of E2 on DLD1 migration was measured after 24 h and 48 h by a wound closure test in control cells and cells cultured in the presence of 1 nM, 10 nM or 100 nM concentrations of E2. Administration of 100 nM E2 significantly decreased migration of DLD1 cells 48 h

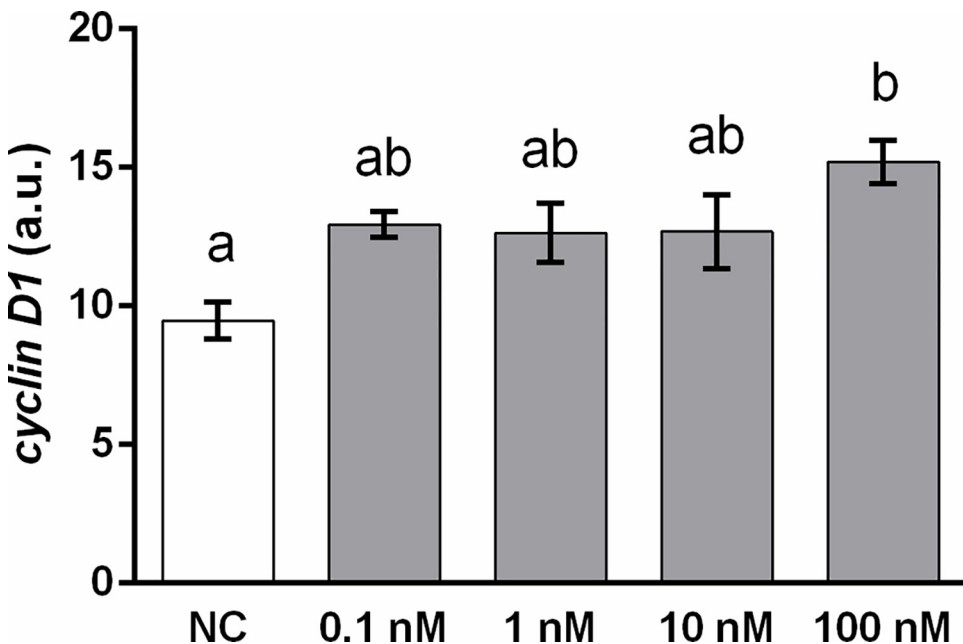

**Fig 4. Expression of *cyclin D1* after a 48-h incubation of DLD1 cells with 17β-estradiol (E2).** E2 was administered in concentrations from 0.1 nM to 100 nM (grey columns). Results are displayed as mean ± SEM (n = 3–4). Statistical differences were determined by one-way ANOVA followed by Tukey's *post hoc* test. Different letters above columns indicate significant differences between groups at the level of $p < 0.05$. NC–negative control (white column).

after wound scratch compared to cells treated with 10 nM E2 (Fig 9). There was a pronounced trend ($p = 0.058$) between control cells and cells treated with 100 nM E2 (Fig 9C).

Next, relative expression of nuclear E2 receptors in DLD1 cells was measured. As is shown in Fig 10, expression of *esr2* is more than 60 times higher than that of *esr1*.

## Discussion

Our results indicate that miR-34a strongly influences the expression of clock genes involved in the circadian basic feedback loop (particularly *per2* and *clock*) without regard for the p53 status of the cell line. The most pronounced effect was observed at the level of *per2* mRNA, as miR-34a administration caused a massive decrease in its expression in DLD1 as well as LoVo cells. In addition to *per2*, miR-34a inhibited the expression of *bmal1* (LoVo, tredn DLD1) and *rev-erbα* (DLD1, trend LoVo) and stimulated the expression of the *clock* gene. miR-34a administration also significantly decreased the levels of *sirt1* (LoVo, DLD1) and *cyclin D1* (LoVo, trend DLD1) mRNA. The response of DLD1 and LoVo cells to miR-34a treatment was similar in most cases. As the majority of human cancers contain a mutated form p53 [57], the effect of miR-34a on cell proliferation and migration was tested using DLD1 cells expressing a truncated form of p53.

miR-34a administration caused a decrease in the proliferation of DLD1 cells, measured by a MTS test, which is in line with the detected decreases in *sirt1* and *cyclin D1* expression. On the other hand, miR-34a administration did not significantly influence wound healing under the present experimental conditions. Unlike miR-34a, E2 administration was able to inhibit proliferation and wound healing in the DLD1 cell line. A synergetic effect of miR-34a and E2 on DLD1 proliferation was not detected.

The inhibitory effect of miR-34a on *per2* expression in DLD1 and LoVo cell lines, as indicated by TargetScan [62], is in line with the results of the present study. Accordingly,

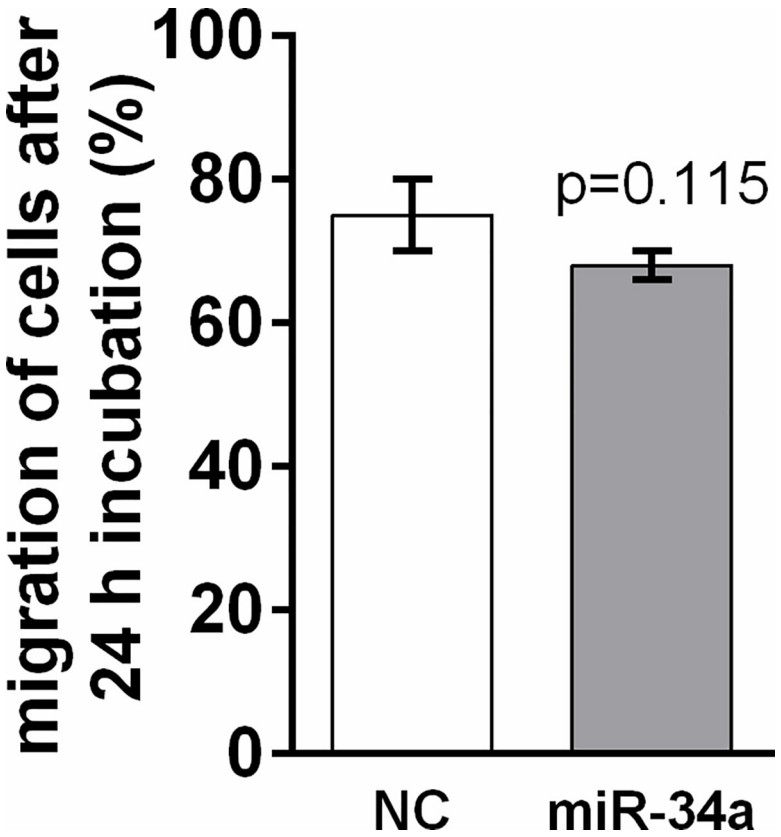

**Fig 5. Migration of DLD1 cells 24 h after miR-34a mimic transfection.** Results are displayed as mean ± SEM (n = 4). Statistical differences were determined by an unpaired Student's t-test. NC–negative control.

administration of mimic miR-34a together with an inhibitor caused an increase in *per2* expression. The same effect has been observed without regard to whether mature miR-34a or pre-miRNA was administered to the DLD1 cell culture [63].

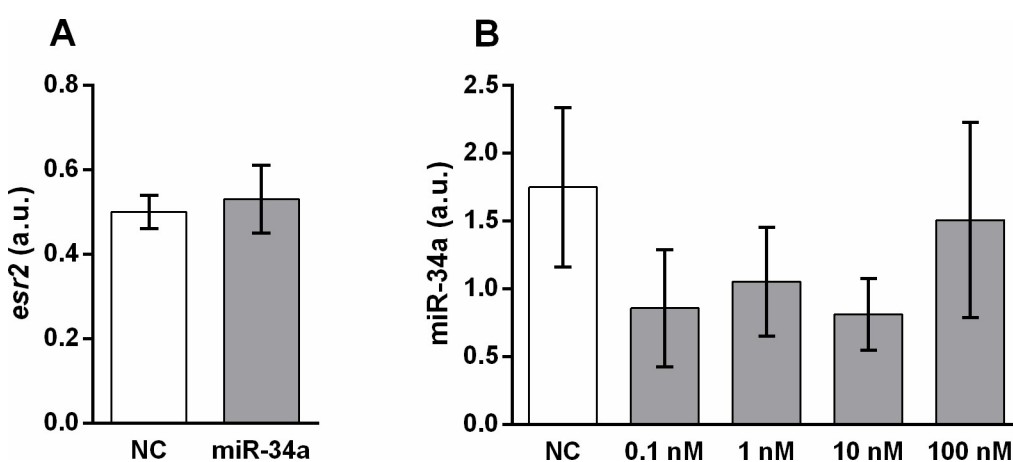

**Fig 6. Appraisal of reciprocal regulation of E2 signalling and miR-34a.** Expression of *esr2* mRNA 48 h after transfection of DLD1 cells with miR-34a mimic or the corresponding control (A). Effect of 17β-estradiol administration on miR-34a expression in the DLD1 cell line (B). Results are displayed as mean ± SEM (n = 3–6). NC–negative control, *esr2* –estrogen receptor 2.

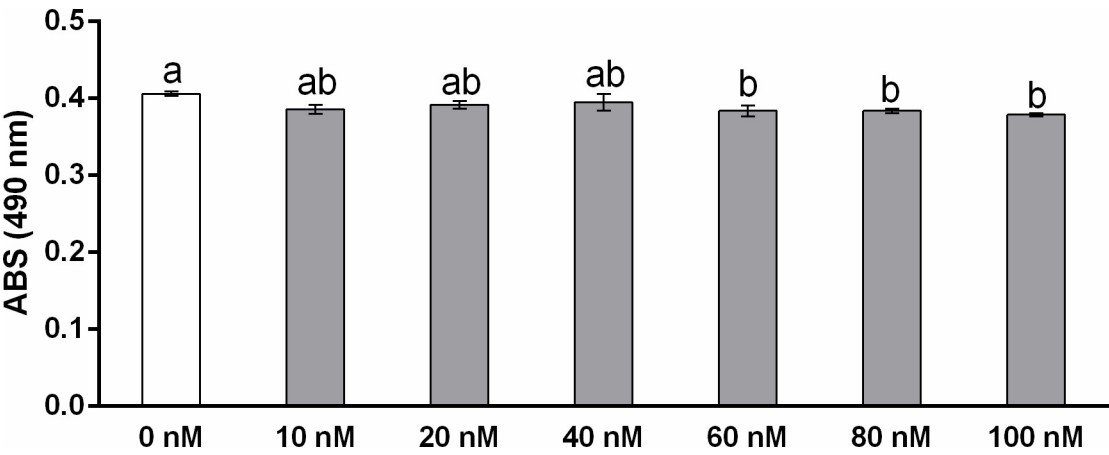

**Fig 7. Proliferation of DLD1 cells after 24 h of incubation with 17β-estradiol (E2) measured by MTS test.** E2 was administered in concentrations of 10, 20, 40, 60, 80 or 100 nM (grey columns). White column indicates control group without E2. Results are displayed as mean ± SEM (n = 6–12). Statistical differences were determined by one-way ANOVA followed by Tukey's *post hoc* test. Different letters above columns indicate significant differences between groups at the level of $p < 0.05$.

In addition to miR-34a, p53 also inhibits *per2* expression. p53's response element overlaps with the E-box and prevents the heterodimer CLOCK/BMAL1 from binding to the *per2* sequence. Under these circumstances, CLOCK/BMAL1-mediated induction of *per2* expression is suppressed [56]. There is also a reciprocal regulatory relationship between p53 and miR-34a functioning, as they potentiate the effects of each other [55,64,65]. To elucidate whether the effect of miR-34a on *per2* expression is direct or p53 mediated, two cell lines differing in p53 status were used in the experiments. DLD1 cells contain a truncated p53 gene

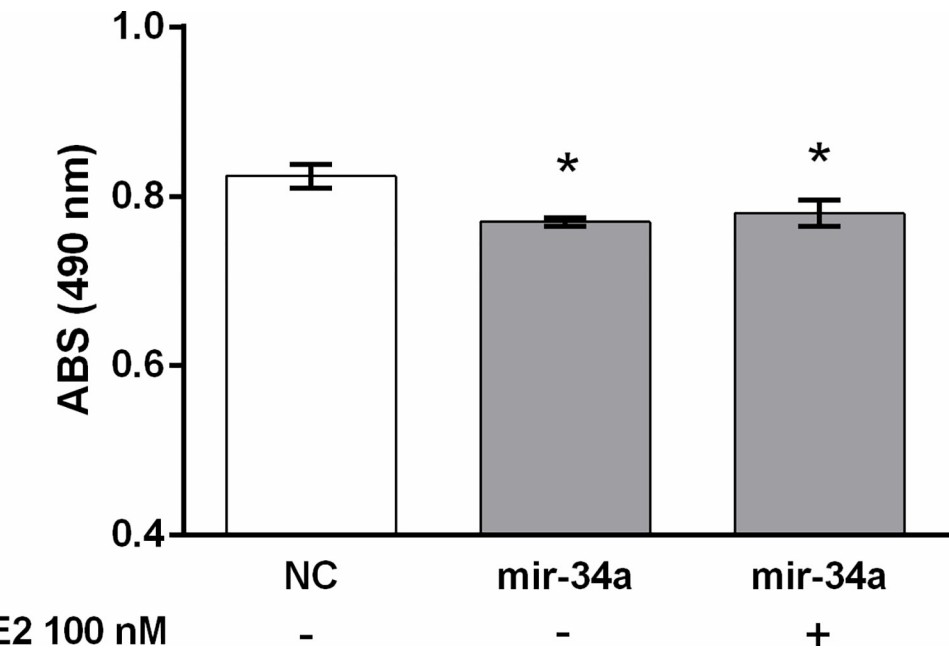

**Fig 8. Proliferation of DLD1 cells measured 48 h after mir-34a transfection with or without parallel E2 administration.** Results are displayed as mean ± SEM (n = 8). Statistical differences were determined by one-way ANOVA followed by Dunnett's multiple comparison test (*$p < 0.05$). NC–negative control, E2–17β estradiol.

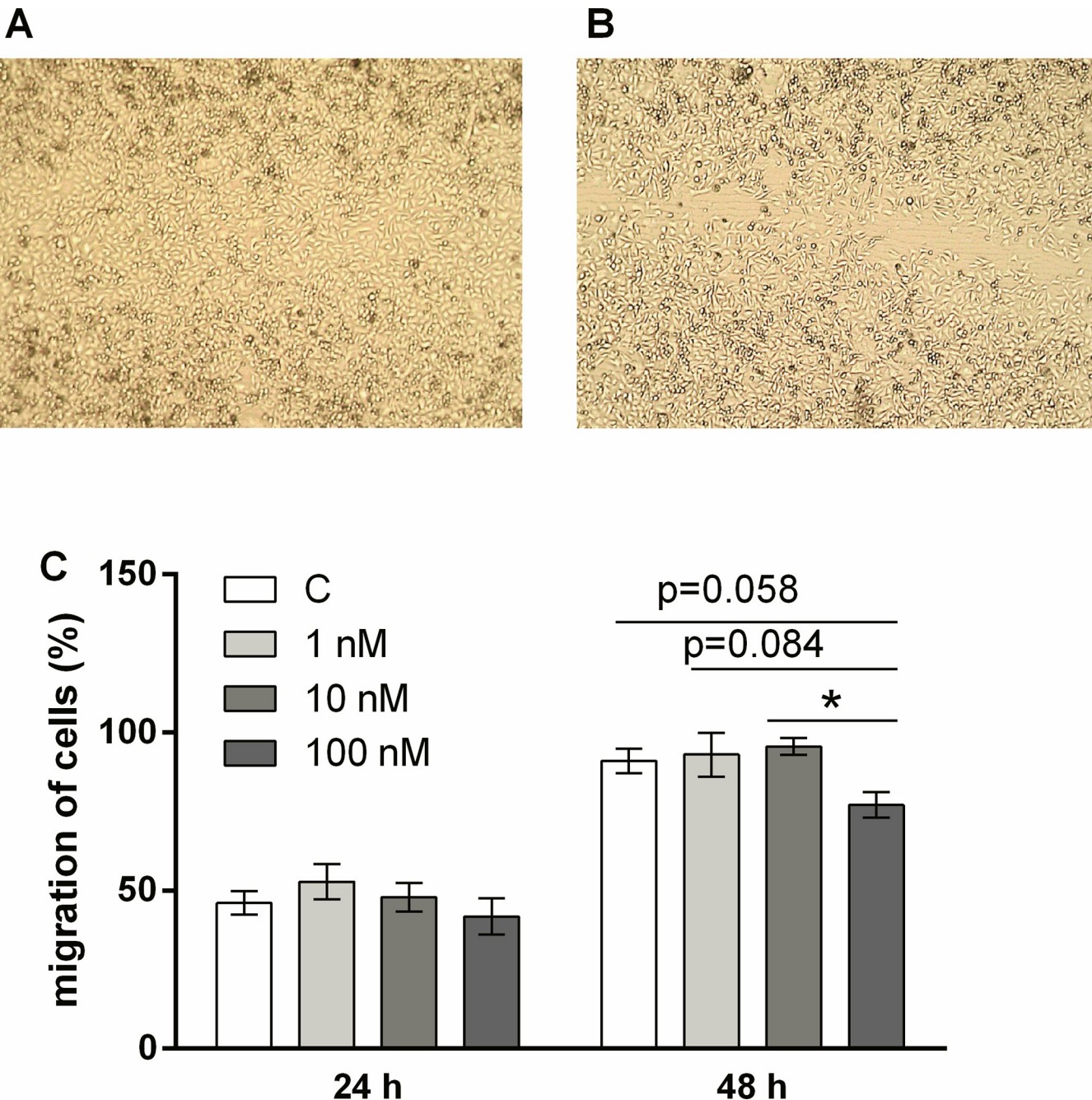

**Fig 9. Migration of DLD1 cells 24 h and 48 h after 17β-estradiol (E2) administration.** Upper panel shows wound closure in control (A) and 100 nM E2-treated DLD1 cells (B, magnification 40x). Lower panel shows the effect of E2 on migration of DLD1 cells after 24 h and 48 h of incubation. Results are displayed as mean ± SEM (n = 4). Statistical differences were determined by one-way ANOVA followed by Tukey's *post hoc* multiple comparison test (* *p* < 0.05). C–control without E2.

that negatively influences p53 binding to its binding domain in the DNA molecule [57], while LoVo cells express fully functional p53. As under the present experimental conditions miR-34a induced a decrease in *per2* expression in DLD1 and LoVo cell lines, we suppose that the effect of miR-34a on *per2* expression does not depend substantially on p53 functionality.

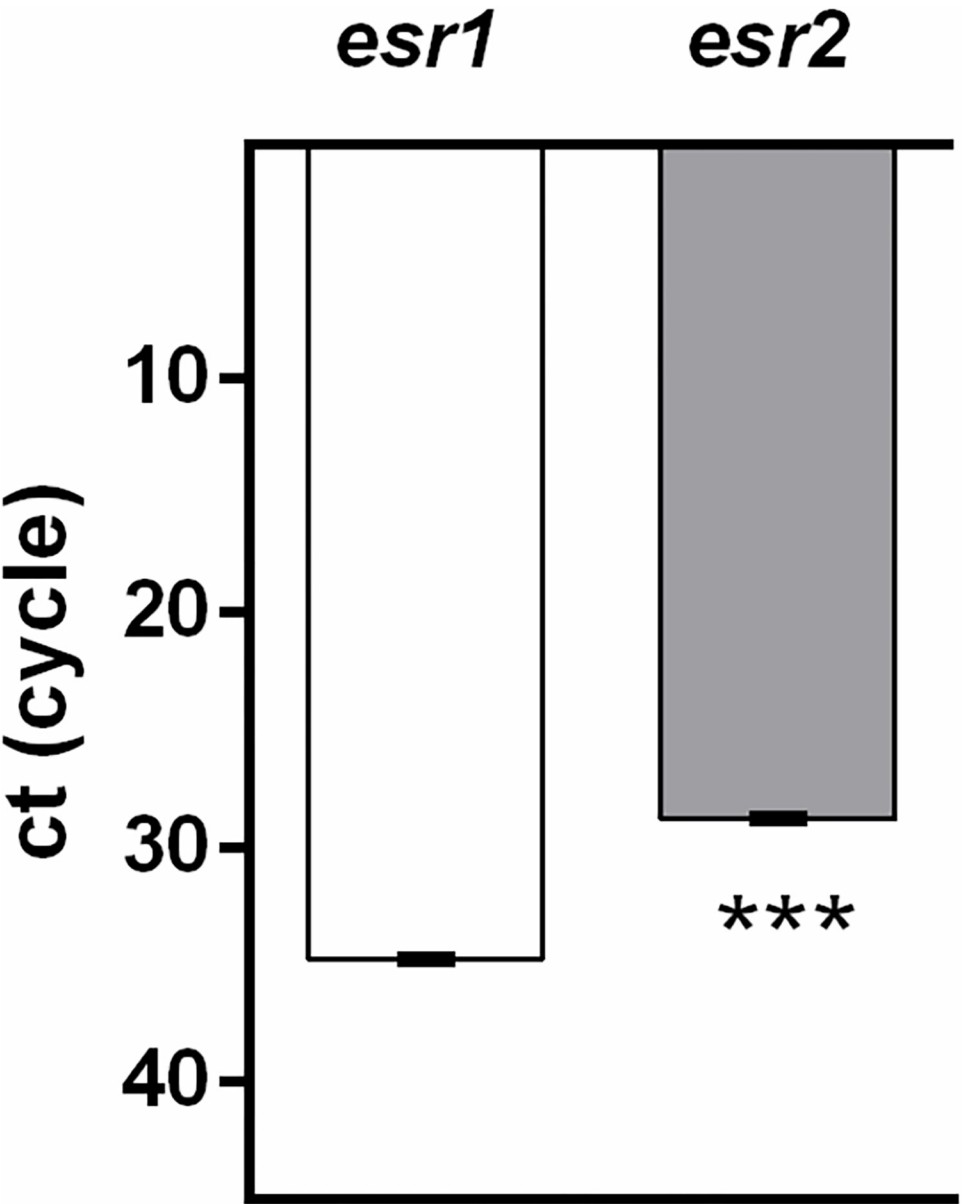

**Fig 10. Expression of *esr1* and *esr2* receptors in the DLD1 cell line.** A cycle threshold (Ct) of 35 was considered the limit of PCR sensitivity in *esr1* measurement. When Ct < 35 or the *esr1* concentration was undetectable, Ct 35 was used for calculation of *esr1* expression. Results are displayed as mean ± SEM (n = 6). Statistical differences were determined by an unpaired Student's t-test (***$p$ < 0.001). *esr1* –estrogen receptor type 1, *esr2* –estrogen receptor type 2.

miR-34a administration also inhibited the expression of the clock gene *bmal1* and the clock-controlled gene *rev-erbα*, which agrees with the search performed in miRWalk [66] and previous research [63]. Surprisingly, we observed an elevated level of the *clock* mRNA concentration after miR-34a administration. It is not currently known if an increase in *clock* expression is a direct effect of miR-34a or a consequence of a deregulated circadian feedback loop. The *clock* gene possesses a miR-34a response element in its 3′-UTR region according to miR-Walk [66]. The stimulatory effect of miRNA on gene expression has been described previously [67]; however, it was related to the 5′-UTR rather than the 3′-UTR [68,69].

In accordance with *in silico* analysis [62], we observed a pronounced decrease in the expression of the clock-controlled genes *sirt1* and *cyclin D1* in the DLD1 and LoVo cell lines. The involvement of miR-34a in regulation of *sirt1* expression is well known [65,70–72]; however, the effect of miR-34a administration on *cyclin D1* expression, to our knowledge, has not been studied previously in colorectal cell lines. miR-34a inhibited *cyclin D1* expression in the non-small-cell lung cancer A549 cell line [73], prostate cancer PC3 cells [74], several oesophageal squamous cell carcinoma cell lines [75], the human breast cancer cell line MCF-7 [76] and the human laryngeal carcinoma cell line HEp-2 [77].

In our study, administration of miR-34a significantly inhibited cell proliferation measured by an MTS assay in DLD1 cells with a mutated form of p53. Inhibitory effects of miR-34a on cell proliferation were previously reported in RKO, HCT116 and LoVo cells [67,78,79]. miR-34a also inhibited proliferation and cell cycle progression in the IMR90 cell line [55] and proliferation and viability in the presence of 5–FU in the HCT8 and SW480 cell lines [80]. Deletion of p53 from the HCT116 genome did not attenuate inhibition of cell proliferation by miR-34a [67].

Sirt1 is likely to be involved in miR-34a-induced lower metabolic activity of DLD1 cells as it participates in regulation of metabolism, inflammation and cell survival [81]. Expression of Sirt1 is frequently upregulated in CRC tissue [82]. It has been shown that downregulation of *sirt1* inhibits proliferation of HCT116 cells [83] and potentiates responsiveness of DLD1 cells to 5-fluorouracil [71].

Interestingly, *per2* is one of numerous targets of *sirt1*. It has been shown that silencing of *sirt1* upregulates and overexpression of *sirt1* causes a decrease in *per2* expression. Similarly, *per2* inhibits expression of *sirt1* via E-box sites located in the *sirt1* promoter [28]. Therefore, there is a transcriptional feedback loop between *sirt1*, *per2* and miR-34a. *sirt1*-mediated inhibition of *per2* can be weakened by miR-34a, which inhibits *sirt1* expression and could lead to an increase in *per2* expression. Despite this, we observed a pronounced decrease in *per2* expression after miR-34a administration. Therefore, we suppose that in the present experimental condition, miR-34a regulation of *per2* overwhelms regulation mediated by *sirt1*.

Inhibition of *cyclin D1* expression in DLD1 cells by miR-34a can also be related to the results of the MTS test. The role of *cyclin D1* as a promotor of cell cycle progression at the G1/S transition in response to mitogenic stimulation is well known [84,85]. A decrease in cell proliferation after *cyclin D1* silencing has been demonstrated previously in ovarian and ependynoma cell lines [86,87]. These observations are in accordance with the actual reported decrease in proliferation and *cyclin D1* expression in DLD1 cells transfected with miR-34a mimic.

Unlike other studies, we only observed a decreasing trend in DLD1 cell migration after miR-34a administration with a mutated form of p53. We suppose that the less potent inhibition of cell migration by miR-34a in our study could have been caused by a combination of two factors. The effect of miR-34a on cell migration is usually studied using cell lines with functional p53 [88–91]. The presence of the point mutation S241F in p53 in the DLD1 cell line [58] that prevents DNA binding [92,93] can influence miR-34a functionality as it has been shown that the inhibitory effect of miR-34a on wound closure was weakened in the HCT116 p53$^{-/-}$ cell line [88]. The concentration of miR-34a mimic used in our experiment was lower (50 nM) than that in other studies using oligos [72,89] and vector delivery of miR-34a [94,95].

miR-34a is generally considered to be a tumour [53,54,96]. However, in accordance with our results, concerns about how supporting evidence about this statement has been gathered recently arose [97]. Although high expression of miR-34a is usually associated with better survival [53,54,60], the opposite association has been reported in one case [98]. The objection was raised that the concentrations of miR-34a used in the experiments demonstrating its tumour suppressor functions were supraphysiological and miR-34a levels in tumour tissues are not

decreased as reported previously [97], which is in accordance with other studies [60,67,98–100]. Similarly, the conclusions of studies where the effect of miR-34a on wound healing were tested are not quite consistent yet and may be context dependent [88,89,94,101].

In the present study, miR-34a inhibited the expression of two genes promoting cell proliferation (*sirt1* and *cyclin D1*); however, it also downregulated two genes (*per2* and *rev-erbα*) with tumour suppressor capacity [40–43,102,103]; respectively). Therefore, the role of miR-34a in CRC progression seems to be more complex than it has been previously declared.

As in our previous study [60], we observed differences in miR-34a expression between male and female patients, and epidemiological evidence demonstrates better CRC survival in female patients compared to males [8], we focused on a possible regulatory relationship between miR-34a and E2. According to others [13,14,22,104,105] and our data [49], E2 mediates its effect in the gastrointestinal tract predominantly via ESR2 receptors that are more abundant in this tissue compared to ESR1 receptors. In the present study, we observed more than 500 times higher expression of ESR2 compared to ESR1 receptors in DLD1 cells. There is a substantial amount of evidence demonstrating that E2 inhibits CRC progression via ESR2 under *in vitro* conditions [22]. The presence of functional ESR2 [18] or ESR2 overexpression [20] was also shown to inhibit CRC progression under *in vivo* conditions in mammals. Protective effects of E2 are, to some extent, lost in CRC tissue as expression of ESR2 is substantially lower in tumours compared to adjacent tissues [106,107].

Expression of *cyclin D1* has been shown to be upregulated via ESR1 receptors and inhibited via ESR2 receptors [13,108]. In the present study, we observed an increase in *cyclin D1* expression when a 100 nM concentration of E2 was administered to DLD1 cells. Despite E2-induced *cyclin D1* expression, E2 inhibited cell migration and proliferation. The revelation of the oncostatic capacity of E2 is in accordance with previously reported results [22]. We suppose that induction of *cyclin D1* expression was mediated by signalling of minorit *esr1* receptors present in DLD1 cells [13,108].

As miR-34a as well as E2 inhibited rate of cell metabolism measured by the MTS assay, we focused on a possible regulatory interaction between miR-34a and E2. However, we did not detect a synergetic effect of miR-34a and E2. Thus, E2 did not influence the expression of miR-34a and miR-34a administration did not influence expression of *esr2* in DLD1 cells. Cyclin D1 seems to be involved in the inhibitory effect of miR-34a on cell proliferation. However, a different mechanism mediated the effect of E2 measured by the MTS test as E2 administration induced *cyclin D1* mRNA expression in DLD1 cells. Therefore, we suppose that sex-dependent differences in miR-34a levels observed in human patients [60] are not related to the E2 concentration.

Actual reported data show that miR-34a influences the expression of several clock genes with the strongest effect on *per2* mRNA expression without respect to p53 status in the examined cell lines. However, when considering the tumour-suppressive effects of miR-34a, this influence seems to be rather counterproductive, as *per2* possesses tumour-suppressor capacity [40,42,43]. Therefore, clock genes should be classified as off-targets of miR-34a when it comes to cancer treatment. Deregulation of the circadian oscillator could be contemplated as a possible negative side effect of miR-34a.

On the other hand, administration of miR-34a-attenuated expression of the metabolic gene *sirt1* and the cell cycle regulator *cyclin D1*, which is in line with its inhibitory influence on DLD1 cell proliferation. E2 co-administration did not modulate the effect of miR-34a on DLD1 proliferation, so for now we cannot support our hypothesis that differences observed in miR-34a expression in CRC patients are caused by differences in E2 levels. We suppose that expression of E2 receptors, which shows stage- and sex-dependent patterns in CRC tumours

[12,109,110], rather than E2 levels, can be related to gender-dependent differences observed in miR-34a expression in human patients.

Among the major limitations of the present study belongs missing evidence about the possible influence of other sex hormones on miR-34a expression in colorectal tumours and CRC cell lines. Based on known data, we indicated clock genes as off-targets of miR-34; however, a positive or negative role of *per2* (or other clock genes) in miR-34a-mediated regulation of DLD1 cell proliferation was not demonstrated. Similarly, it cannot be concluded to which extent changes in the expression of miR-34a-responsive clock-controlled genes are influenced by a deregulated circadian feedback loop.

As during cancer progression, the levels of both ESR2 and miR-34a are modified in tumour tissue, in future it would be of interest to test all studied regulatory relationships in non-cancerous colon cell lines. We believe that research should also be focused on the functional relationship between miR-34a and the circadian oscillator in cancer and non-cancerous cells. It would be very helpful if epidemiological research is focused more on sex hormone receptors in the colon and rectum in healthy probands during the whole life span. Improved knowledge about E2 signalling in these tissues over time would help to determine if E2 signalling contributes to the sex-dependent changes observed in CRC patients.

## Conclusions

To conclude, miR-34a strongly influences the expression of clock (*per2*, *bmal1* and *clock*) and clock controlled (*rev-erbα*, *sirt1* and *cyclin D1*) genes. Ectopic mir-34a inhibits the expression of *per2* without respect to p53 status, but its oncostatic effects are more likely caused by inhibition of *sirt1* and *cyclin D1* expression. E2 administration inhibits the growth of DLD1 cells; however, this effect seems to be independent of miR-34a-mediated action. With respect to the possible use of miR-34a in cancer treatment, *per2* can be considered as an off-target gene. Its possibly ambiguous oncogenic character should be taken into consideration in future clinical studies focused on miR-34a.

## Supporting information

**S1 Fig. Expression of miR-34a in control and miR-34a mimic-transfected DLD1 cells.** Results are displayed as mean ± SEM (n = 5–6). Statistical differences were determined by a Student's unpaired t-test (\*\*\*$p < 0.001$). NC–negative control. (TIF)

**S2 Fig.** Fluorescent staining of PER2 protein in control (A) and pre-miR-34a-transfected DLD1 cells (B). Fluorescence intensity (C) is expressed as mean ± SEM (n = 12). Images were taken with an inverted fluorescence microscope with a magnification of 100x. Statistical differences were determined by an unpaired Student's t-test (\*$p < 0.05$). preNC–negative control. (TIF)

**S1 Table. Sequences of the primers used in real-time polymerase chain reaction and reverse transcription.** (DOC)

## Author Contributions

**Conceptualization:** Iveta Herichová.

**Formal analysis:** Roman Moravčík, Soňa Olejárová, Iveta Herichová.

**Funding acquisition:** Iveta Herichová.

**Investigation:** Roman Moravčík, Soňa Olejárová, Iveta Herichová.

**Methodology:** Roman Moravčík, Soňa Olejárová, Iveta Herichová.

**Project administration:** Iveta Herichová.

**Resources:** Iveta Herichová.

**Supervision:** Iveta Herichová.

**Validation:** Soňa Olejárová.

**Visualization:** Soňa Olejárová.

**Writing – original draft:** Roman Moravčík, Jana Zlacká, Iveta Herichová.

**Writing – review & editing:** Roman Moravčík, Jana Zlacká, Iveta Herichová.

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
