## [Decision Letter · Decision Letter 0]

20 Jul 2023

PONE-D-23-17825Effect of miR-34a on expression of clock and clock-controlled genes in human cancer DLD1 and Lovo cells

and 17β-estradiol-mediated regulationPLOS ONE

Dear Dr. Herichova,

Thank you for submitting your manuscript to PLOS ONE. After careful consideration, we feel that it has merit but does not fully meet PLOS ONE’s publication criteria as it currently stands. Therefore, we invite you to submit a revised version of the manuscript that addresses the points raised during the review process.

We look forward to receiving your revised manuscript.

Kind regards,

Shiki Fujino, M.D.

Academic Editor

PLOS ONE

Journal Requirements:

"This research was supported by grants APVV-16-0209, APVV-20-0241 and VEGA 1/0455/23."

"IH

The Slovak Research and Development Agency

https://www.apvv.sk/?lang=en

APVV-16-0209

APVV-20-0241

Scientific Grant Agency of the Ministry of Education, Science, Research and Sport of the Slovak Republic

https://www.minedu.sk/vedecka-grantova-agentura-msvvas-sr-a-sav-vega/

VEGA 1/0455/23

Additional Editor Comments:

Please read reviewer's comments carefully, and respond to them.

Reviewers' comments:

Reviewer's Responses to Questions

**Comments to the Author**

1. Is the manuscript technically sound, and do the data support the conclusions?

Reviewer #1: Yes

Reviewer #2: Yes

Reviewer #3: Partly

2. Has the statistical analysis been performed appropriately and rigorously? 

Reviewer #1: Yes

Reviewer #2: Yes

Reviewer #3: Yes

3. Have the authors made all data underlying the findings in their manuscript fully available?

Reviewer #1: Yes

Reviewer #2: Yes

Reviewer #3: Yes

4. Is the manuscript presented in an intelligible fashion and written in standard English?

Reviewer #1: Yes

Reviewer #2: Yes

Reviewer #3: Yes

5. Review Comments to the Author

Reviewer #1: Dear authors, I read with much interest your work.

I was surprised that the title does not resemble the aim of the study.

Line 404

Which previous study? "the results of our PREVIOUS study, where administration of miR-34a in

100nM concentration suppressed per2 expression in the DLD1 cell line" I suggest to change "previous" with "in the present study"

the same for " As under recent experimental conditions ......suggest changing "recent" for "the present"

The same for line 427; "This finding is in agreement with our previous study

and "with a recently reported decrease in proliferation and cyclin D1......" I suggest "with the actual reported......"

Line 541, again;

It is difficult to understand when you refer to what you saw in the present study with what you saw in another previous study

Line 548, again

Line 560, again

554. change "similary" for "Thus"

561, change "respect to p53 status of the examined cell lines" change "of" with "IN the examined..."

In line 53, you are missing a "be"

Line 389, I suppose you meant "our" results...

Line 470; please control this sentence: Interestingly, per2 is one of numerous targets of sirt1. It was shown that silencing sirt1 expression causes an increase and overexpression of sirt1 INHIBITS per2 expression."

Reviewer #2: Dear Authors,

There are some comments on the manuscript:

i) The Introduction Section is overweighted. Please, make this section more structurized and precised.

ii) Please, provide information about the Y chromosome presence in the investigated cell lines. If these cell lines had different Y chromosome status you should discuss this issue.

iii) The Discussing Section is complicated. Please, make this section more scientifically sounded. Provide the limitations for this study and prognosis for the future investigations.

Reviewer #3: 1) The introduction section seems to be a little too long.

I suggest removing 94-99 lines; 110- 115 lines; 135-149 lines and 163-172 lines as over excessive information. These paragraphs are perfect for the review articles, but in this case are too excessively explanatory.

Paragraph (line 100-109) could be shortened and combined with paragraph in lines 116-125.

Paragraph (line 150-162) could be shortened (by removing lines 153-162, which review canonical miRNA maturation pathway/ unspecific application as biomarkers) and combined with paragraph in lines 173-180.

The present introduction doesn’t describe the basic novelty, experimental design and main results of the present study. Please insert an additional paragraph.

2) The method section is written as a whole text. Please discriminate distinct methods as separate.

3) Result section is poorly organized. Some of the figures can be combined (e.g. figure 1-4). Some of the figures could be presented as supplementary (e.g. figure 1).

Quality of data is questionable:

The authors present mRNA data of selected genes and a significant change in expression following miRNA mimic transfection. Why the authors didn’t perform western blot analysis to validate such results as miRNAs in mammalian cells target mRNA translation (which leads to decreased protein levels), not degradation (changes in mRNA levels).

The quality of Fluorescence microscopy images is very poor (figure 3A). How did the authors normalize protein fluorescence data points between samples?

Questionable experimental design:

The authors measure mRNA levels at different times (24 and 48h). Why such timelines where selected?

The present study lack of any conclusions!!!!

6. PLOS authors have the option to publish the peer review history of their article (what does this mean?). If published, this will include your full peer review and any attached files.

Reviewer #1: No

Reviewer #2: No

Reviewer #3: No

---

## [Author Response · Author response to Decision Letter 0]

14 Aug 2023

Reviewer #1:

Dear reviewer,

Thank you very much for your valuable time, suggestions and advice. All your suggestions and comments were implemented into the text, please see detailed response bellow. However, on request Reviewer #2 section “Discussion” was rewritten so extensively, that in some cases whole sentences, including some of those that you have mentioned in your comments, had to be deleted. 

We hope that text of MS was improved in such a way that it is acceptable for publication now. 

With many thanks and best regards,

the authors

Dear authors, I read with much interest your work.

I was surprised that the title does not resemble the aim of the study.

- thank you for the suggestion, the title was changed to reflect better aims of the study

Line 404

Which previous study? "the results of our PREVIOUS study, where administration of miR-34a in

100nM concentration suppressed per2 expression in the DLD1 cell line" I suggest to change "previous" with "in the present study"

- thank you very much for the comment, indicated paragraph was changed according to you advice.

the same for " As under recent experimental conditions ......suggest changing "recent" for "the present"

The same for line 427; "This finding is in agreement with our previous study

and "with a recently reported decrease in proliferation and cyclin D1......" I suggest "with the actual reported......"

Line 541, again;

- thank you very much for the comment, text was changed according to your advice, some sentences were completely omitted in the revised version of MS as Reviewer #2 required to make discussion more compact.

It is difficult to understand when you refer to what you saw in the present study with what you saw in another previous study

Line 548, again

Line 560, again

- thank you very much for the comment, indicated paragraph was rewritten to make content more coherent

554. change "similary" for "Thus"

- thank you very much for the comment, text was changed according to your advice

561, change "respect to p53 status of the examined cell lines" change "of" with "IN the examined..." In line 53, you are missing a "be"

- thank you very much, text was corrected accordingly

Line 389, I suppose you meant "our" results...

- yes, thank you, text was corrected

Line 470; please control this sentence: Interestingly, per2 is one of numerous targets of sirt1. It was shown that silencing sirt1 expression causes an increase and overexpression of sirt1 INHIBITS per2 expression."

- thank you very much for the comment, sentence was corrected

Reviewer #2: 

Dear reviewer,

Thank you very much for your valuable time and suggestions, we appreciate them very much. All your suggestions were implemented into the text. Concern about possible difference in Y chromosome status was explained, please see detailed response bellow. We hope that you will find revised MS acceptable for publication. 

With many thanks and best regards,

the authors

Dear Authors,

There are some comments on the manuscript:

i) The Introduction Section is overweighted. Please, make this section more structurized and precised.

- thank you very much for the comment. Section „Introduction“ was shortened substantially. In particular, lines 94-99, 110- 115, 135-149 and 163-172 were removed. Paragraph in lines 100-109 was shortened and combined with paragraph in lines 116-125. Paragraph in lines 150-162 was also shortened by removing lines 153-162 and combined with paragraph in lines 173-180. We included short paragraph explaining novelty, experimental design and main results as it was requested by Reviewer #3. Recent version of introduction contains 1014 words instead of 1631 words. We believe that revised introduction is sufficiently focused and compact.

ii) Please, provide information about the Y chromosome presence in the investigated cell lines. If these cell lines had different Y chromosome status you should discuss this issue.

- thank you very much for this idea. DLD1 and LoVo cell lines are derived from male patients diagnosed with colon adenocarcinoma. These cell lines are considered to be cytogenetically similar. According to Knutsen et al. (2010) and Abdel-Rahman et al. (2001) they are nearly-diploid cell lines with some structural abnormalities in some of their autosome chromosomes, however, both cell lines possess Y chromosome so they do not differ in this respect.

Knutsen T, Padilla-Nash HM, Wangsa D, Barenboim-Stapleton L, Camps J, McNeil N, Difilippantonio MJ, Ried T. Definitive molecular cytogenetic characterization of 15 colorectal cancer cell lines. Genes Chromosomes Cancer. 2010; 49(3):204-23. doi: 10.1002/gcc.20730.

https://onlinelibrary.wiley.com/doi/epdf/10.1002/gcc.20730

Abdel-Rahman WM, Katsura K, Rens W, Gorman PA, Sheer D, Bicknell D, Bodmer WF, Arends MJ, Wyllie AH, Edwards PA. Spectral karyotyping suggests additional subsets of colorectal cancers characterized by pattern of chromosome rearrangement. Proc Natl Acad Sci U S A. 2001; 98(5):2538-43. doi: 10.1073/pnas.041603298.

https://www.pnas.org/doi/epdf/10.1073/pnas.041603298

iii) The Discussing Section is complicated. Please, make this section more scientifically sounded. Provide the limitations for this study and prognosis for the future investigations.

- thank you for the comment, section „Discussion“ was rewritten to make it more concise and focused. We incorporated paragraphs devoted to limitations of the study and prognosis of research. On request Reviewer #3 we also incorporated paragraph “Conclusions” at the end of section “Discussion”.

Reviewer #3: 

Dear reviewer,

Thank you very much for your valuable time, suggestions, constructive comments, advice and questions, we appreciate them very much. All your suggestions and comments were implemented into the text and questions were answered, please see detailed response bellow. We hope that you will find revised MS acceptable for publication. 

With many thanks and best regards,

the authors

1) The introduction section seems to be a little too long.

I suggest removing 94-99 lines; 110- 115 lines; 135-149 lines and 163-172 lines as over excessive information. These paragraphs are perfect for the review articles, but in this case are too excessively explanatory.

Paragraph (line 100-109) could be shortened and combined with paragraph in lines 116-125.

Paragraph (line 150-162) could be shortened (by removing lines 153-162, which review canonical miRNA maturation pathway/ unspecific application as biomarkers) and combined with paragraph in lines 173-180.

The present introduction doesn’t describe the basic novelty, experimental design and main results of the present study. Please insert an additional paragraph.

- dear reviewer, thank you very much for detailed advice how to reorganize section „Introduction“. We implemented all your suggestion. Additional two paragraphs focused on novelty, experimental design and results were added at the end of introduction. We believe that revised introduction is sufficiently focused and compact.

2) The method section is written as a whole text. Please discriminate distinct methods as separate.

- thank you very much for the suggestion. In revised form manuscript section „Material and methods“ is separated into sub-sections:

• Cell lines

• miRNA used in the study and their transfection

• RNA isolation and PCR

• Fluorescent staining of PER2

• Evaluation of the metabolic activity and viability of cells by MTS test

- Effect of E2

- Combined effect of E2 and miR-34a

• Wound healing assay

• Statistical analysis 

3) Result section is poorly organized. Some of the figures can be combined (e.g. figure 1-4). Some of the figures could be presented as supplementary (e.g. figure 1).

- thank you for the suggestion. Figure 1 was incorporated in revised version of MS as supplementary Figure 1. Former figure 3 was improved and in revised version of MS is provided as supplementary Figure 2. Former figures 2 and 4 were joined and in the revised version of MS are incorporated as Figure 1. Numbering of all other figures was shifted accordingly: Fig. 5 to Fig. 2, Fig. 6 go Fig. 3 etc.

Quality of data is questionable:

The authors present mRNA data of selected genes and a significant change in expression following miRNA mimic transfection. Why the authors didn’t perform western blot analysis to validate such results as miRNAs in mammalian cells target mRNA translation (which leads to decreased protein levels), not degradation (changes in mRNA levels).

- we agree that inhibitory influence of miRNA always issue into decrease of translation. The first studies really indicated that in miRNAs inhibit gene expression mainly by influencing translation. However, since that more complex miRNA mediated regulation of transcriptome was revealed and mRNA degradation in response to miRNA treatment was firmly proved (e.g. Guo et al., 2010, 3121 references in WOS). Later dominant role of mRNA degradation in execution of miRNA mediated effects was described in detail (e.g. Huntzinger and Izaurralde, 2011, 1792 references in WOS). Recently it is well accepted that miRNA mediated inhibition of proteosynthesis includes deadenylation and decaping of mRNA and consequent degradation of target mRNA (e.g. O'Brien et al., 2018, 2192 references in WOS).

Guo H, Ingolia NT, Weissman JS, Bartel DP. Mammalian microRNAs predominantly act to decrease target mRNA levels. Nature. 2010; 466(7308):835-40. doi: 10.1038/nature09267.

https://pubmed.ncbi.nlm.nih.gov/20703300/

Huntzinger E, Izaurralde E. Gene silencing by microRNAs: contributions of translational repression and mRNA decay. Nat Rev Genet. 2011; 12(2):99-110. doi: 10.1038/nrg2936. 

https://pubmed.ncbi.nlm.nih.gov/21245828/

O'Brien J, Hayder H, Zayed Y, Peng C. Overview of MicroRNA Biogenesis, Mechanisms of Actions, and Circulation. Front Endocrinol (Lausanne). 2018; 9:402. doi: 10.3389/fendo.2018.00402.

https://pubmed.ncbi.nlm.nih.gov/30123182/

Based on finding that miRNAs influence mRNA levels “The experimentally validated microRNA-target interactions database” (miRTarBase) accepts qPCR as strong evidence of miRNA and target gene interaction.

https://mirtarbase.cuhk.edu.cn/~miRTarBase/miRTarBase_2022/php/index.php

e.g. interaction of miR-34a and Sirt1

 Validation methods

 Strong evidence Less strong evidence

ID Species (miRNA) Species (Target) miRNA Target Reporter assay Western blot qPCR Microarray NGS pSILAC Other CLIP-Seq Sum Number of paper

MIRT733820 hsa hsa hsa-miR-34a-3p SIRT1 0 0 1 0 0 0 0 0 1 0

MIRT002098 hsa hsa hsa-miR-34a-5p SIRT1 1 1 1 0 0 0 1 0 4 18

MIRT003150 hsa rno hsa-miR-34a-5p Sirt1 0 1 1 1 0 0 0 0 3 1

miRTarBase is highly reputable database as it is indicated by 4333 references in WOS (sum for all updates):

Huang et al.: miRTarBase update 2022: an informative resource for experimentally validated miRNA–target interactions. Nucleic Acids Research 2022, 50(D1):D222-D230.

- 96 citations in WOS

Huang et al.: miRTarBase 2020: updates to the experimentally validated microRNA-target interaction database. Nucleic Acids Research 2020, 48(D1):D148-D154.

- 709 citations in WOS

Chou et al.: miRTarBase update 2018: A resource for experimentally validated microRNA-target interactions. Nucleic Acids Research 2018, 46(D1):D296-D302.

- 1233 citations in WOS

Chou et al.: miRTarBase 2016: updates to the experimentally validated miRNA-target interactions database. Nucleic Acids Research 2016, 44(D1):D239-47.

- 803 citations in WOS

Hsu et al.: miRTarBase update 2014: an information resource for experimentally validated miRNA-target interactions. Nucleic Acids Research 2014, 42(Database issue):D78-85.

- 566 citations in WOS

Hsu et al.: miRTarBase: a database curates experimentally validated microRNA-target interactions" Nucleic Acids Research 2011, 39(Database issue):D163-9.

- 926 citations in WOS

Regulation of clock genes occurs mainly at the transcriptional level via E-box in their promoter. Basic feedback loop controls transcription of so called clock controlled genes also via E-box (Reppert and Weaver, 2001). It is known that pronounced changes in clock gene expression are followed by change in corresponding protein level and therefore results based on measurement of clock gene mRNA expression are well accepted in biological rhythm oriented research (e.g. Yan et al., 1999; Oishi et al., 1998). Bellow we provide list of paper for all genes mentioned in recent MS showing, that expression of proteins follows expression of mRNA. 

Reppert SM, Weaver DR. Molecular analysis of mammalian circadian rhythms. Annu Rev Physiol. 2001; 63:647-76. doi: 10.1146/annurev.physiol.63.1.647.

- 1167 citations in WOS

Yan L, Takekida S, Shigeyoshi Y, Okamura H. Per1 and Per2 gene expression in the rat suprachiasmatic nucleus: circadian profile and the compartment-specific response to light. Neuroscience. 1999; 94(1):141-50. doi: 10.1016/s0306-4522(99)00223-7.

- 206 citations in WOS

Oishi K, Sakamoto K, Okada T, Nagase T, Ishida N. Antiphase circadian expression between BMAL1 and period homologue mRNA in the suprachiasmatic nucleus and peripheral tissues of rats. Biochem Biophys Res Commun. 1998; 253(2):199-203. doi: 10.1006/bbrc.1998.9779.

-191 citations in WOS

List of papers demonstrating positive correlation between mRNA and protein concentration of genes measured in MS:

PER2

Xiong H, Yang Y, Yang K, Zhao D, Tang H, Ran X. Loss of the clock gene PER2 is associated with cancer development and altered expression of important tumor-related genes in oral cancer. Int J Oncol. 2018; 52(1):279-287. doi: 10.3892/ijo.2017.4180. 

https://www.spandidos-publications.com/10.3892/ijo.2017.4180

Li YY, Jin F, Zhou JJ, Yu F, Duan XF, He XY, Wang R, Wu WL, Long JH, Luo XL. Downregulation of the circadian Period family genes is positively correlated with poor head and neck squamous cell carcinoma prognosis. Chronobiol Int. 2019; 36(12):1723-1732. doi: 10.1080/07420528.2019.1648486.

https://www.tandfonline.com/doi/abs/10.1080/07420528.2019.1648486?journalCode=icbi20

Wang X, Wang L, Yu Q, Xu Y, Zhang L, Zhao X, Cao X, Li Y, Li L. Alterations in the expression of Per1 and Per2 induced by Aβ31-35 in the suprachiasmatic nucleus, hippocampus, and heart of C57BL/6 mouse. Brain Res. 2016; 1642:51-58. doi: 10.1016/j.brainres.2016.03.026.

https://www.sciencedirect.com/science/article/pii/S0006899316301573?via%3Dihub

BMAL1

Asher G, Gatfield D, Stratmann M, Reinke H, Dibner C, Kreppel F, Mostoslavsky R, Alt FW, Schibler U. SIRT1 regulates circadian clock gene expression through PER2 deacetylation. Cell. 2008; 134(2):317-28. doi: 10.1016/j.cell.2008.06.050. 

https://www.cell.com/action/showPdf?pii=S0092-8674%2808%2900837-4

Cha S, Wang J, Lee SM, Tan Z, Zhao Q, Bai D. Clock-modified mesenchymal stromal cells therapy rescues molecular circadian oscillation and age-related bone loss via miR142-3p/Bmal1/YAP signaling axis. Cell Death Discov. 2022; 8(1):111. doi: 10.1038/s41420-022-00908-7. https://www.nature.com/articles/s41420-022-00908-7

Liu S, Zhou Y, Chen Y, Liu Y, Peng S, Cao Z, Xia H. Bmal1 promotes cementoblast differentiation and cementum mineralization via Wnt/β-catenin signaling. Acta Histochem. 2022; 124(3):151868. doi: 10.1016/j.acthis.2022.151868. 

https://www.sciencedirect.com/science/article/pii/S0065128122000277?via%3Dihub

ESR2

Wróbel AM, Gregoraszczuk EŁ. Actions of methyl-, propyl- and butylparaben on estrogen receptor-α and -β and the progesterone receptor in MCF-7 cancer cells and non-cancerous MCF-10A cells. Toxicol Lett. 2014; 230(3):375-81. doi: 10.1016/j.toxlet.2014.08.012.

https://www.sciencedirect.com/science/article/pii/S0378427414012995?via%3Dihub

Xue Q, Lin Z, Cheng YH, Huang CC, Marsh E, Yin P, Milad MP, Confino E, Reierstad S, Innes J, Bulun SE. Promoter methylation regulates estrogen receptor 2 in human endometrium and endometriosis. Biol Reprod. 2007; 77(4):681-7. doi: 10.1095/biolreprod.107.061804.

https://www.webofscience.com/wos/woscc/full-record/WOS:000249766700011

Hojnik M, Sinreih M, Anko M, Hevir-Kene N, Knific T, Pirš B, Grazio SF, Rižner TL. The Co-Expression of Estrogen Receptors ERα, ERβ, and GPER in Endometrial Cancer. Int J Mol Sci. 2023; 24(3):3009. doi: 10.3390/ijms24033009.

https://www.mdpi.com/1422-0067/24/3/3009

CLOCK

Lengyel Z, Lovig C, Kommedal S, Keszthelyi R, Szekeres G, Battyáni Z, Csernus V, Nagy AD. Altered expression patterns of clock gene mRNAs and clock proteins in human skin tumors. Tumour Biol. 2013 Apr;34(2):811-9. doi: 10.1007/s13277-012-0611-0.

https://link.springer.com/article/10.1007/s13277-012-0611-0

Jiang P, Xu C, Zhang P, Ren J, Mageed F, Wu X, Chen L, Zeb F, Feng Q, Li S. Epigallocatechin‑3‑gallate inhibits self‑renewal ability of lung cancer stem‑like cells through inhibition of CLOCK. Int J Mol Med. 2020 Dec;46(6):2216-2224. doi: 10.3892/ijmm.2020.4758.

https://www.spandidos-publications.com/10.3892/ijmm.2020.4758

Xu H, Wang Z, Mo G, Chen H. Association between circadian gene CLOCK and cisplatin resistance in ovarian cancer cells: A preliminary study. Oncol Lett. 2018, 15(6):8945-8950. doi: 10.3892/ol.2018.8488.

https://www.spandidos-publications.com/10.3892/ol.2018.8488

REV-ERBA

Chawla A, Lazar MA. Induction of Rev-ErbA alpha, an orphan receptor encoded on the opposite strand of the alpha-thyroid hormone receptor gene, during adipocyte differentiation. J Biol Chem. 1993; 268(22):16265-9.

https://www.jbc.org/article/S0021-9258(19)85415-7/fulltext

Pinto AP, Muñoz VR, da Rocha AL, Rovina RL, Ferrari GD, Alberici LC, Simabuco FM, Teixeira GR, Pauli JR, de Moura LP, Cintra DE, Ropelle ER, Freitas EC, Rivas DA, da Silva ASR. IL-6 deletion decreased REV-ERBα protein and influenced autophagy and mitochondrial markers in the skeletal muscle after acute exercise. Front Immunol. 2022; 13:953272. doi: 10.3389/fimmu.2022.953272.

https://www.frontiersin.org/articles/10.3389/fimmu.2022.953272/full

SIRT1

Zhao X, Jin Y, Yang L, Hou Z, Liu Y, Sun T, Pei J, Li J, Yao C, Wang X, Chen G. Promotion of SIRT1 protein degradation and lower SIRT1 gene expression via reactive oxygen species is involved in Sb-induced apoptosis in BEAS-2b cells. Toxicol Lett. 2018; 296:73-81. doi: 10.1016/j.toxlet.2018.07.047.

https://www.sciencedirect.com/science/article/pii/S0378427418315340?via%3Dihub

Li B, Hu Y, Li X, Jin G, Chen X, Chen G, Chen Y, Huang S, Liao W, Liao Y, Teng Z, Bin J. Sirt1 Antisense Long Noncoding RNA Promotes Cardiomyocyte Proliferation by Enhancing the Stability of Sirt1. J Am Heart Assoc. 2018; 7(21):e009700. doi: 10.1161/JAHA.118.009700.

https://www.ahajournals.org/doi/epub/10.1161/JAHA.118.009700

Jin Q, Zhang F, Yan T, Liu Z, Wang C, Ge X, Zhai Q. C/EBPalpha regulates SIRT1 expression during adipogenesis. Cell Res. 2010; 20(4):470-9. doi: 10.1038/cr.2010.24.

https://www.nature.com/articles/cr201024

CYCLIN D1

Maxwell M, Galanopoulos T, Antoniades H. Cell-cycle regulator cyclin D1 mRNA and protein overexpression occurs in primary malignant gliomas. Int J Oncol. 1996; 9(3):493-7. doi: 10.3892/ijo.9.3.493.

https://www.spandidos-publications.com/ijo/9/3/493

Li X, Tian Z, Jin H, Xu J, Hua X, Yan H, Liufu H, Wang J, Li J, Zhu J, Huang H, Huang C. Decreased c-Myc mRNA Stability via the MicroRNA 141-3p/AUF1 Axis Is Crucial for p63α Inhibition of Cyclin D1 Gene Transcription and Bladder Cancer Cell Tumorigenicity. Mol Cell Biol. 2018; 38(21):e00273-18. doi: 10.1128/MCB.00273-18.

https://www.tandfonline.com/doi/full/10.1128/MCB.00273-18?scroll=top&needAccess=true&role=tab

Xiao D, Chinnappan D, Pestell R, Albanese C, Weber HC. Bombesin regulates cyclin D1 expression through the early growth response protein Egr-1 in prostate cancer cells. Cancer Res. 2005; 65(21):9934-42. doi: 10.1158/0008-5472.CAN-05-1830.

https://aacrjournals.org/cancerres/article/65/21/9934/657416/Bombesin-Regulates-Cyclin-D1-Expression-through

Lin S, Wang W, Wilson GM, Yang X, Brewer G, Holbrook NJ, Gorospe M. Down-regulation of cyclin D1 expression by prostaglandin A(2) is mediated by enhanced cyclin D1 mRNA turnover. Mol Cell Biol. 2000; 20(21):7903-13. doi: 10.1128/MCB.20.21.7903-7913.2000.

https://www.tandfonline.com/doi/full/10.1128/MCB.20.21.7903-7913.2000?scroll=top&needAccess=true&role=tab

We preferred fluorescent staining of PER2 before western blot as it includes much less manipulation with cells resp. samples. Fluorescent staining is performed in situ on slide where cells are cultured all the time. In this way we were able to eliminate a lot of possible mistakes coming from more complicated Western blot.

The quality of Fluorescence microscopy images is very poor (figure 3A). How did the authors normalize protein fluorescence data points between samples?

- Images of fluorescent PER2 staining with better resolution are provided in revised version of MS. Former Figure 3 is in revised version of MS presented as supplementary Figure 2. 

- image analysis and normalisation was performed with use of Image J software. This information is now mentioned in the section „Materials and methods, Statistical analysis“. To eliminate possible differences caused by non-specific staining in miR-34a and negative control treated cells were analysed on one glass slide with the same batch of solutions and antibodies. All methodological steps during analysis occurred at the same time. After taking pictures, the same threshold was set in figures showing effect of miR-34a and negative control. There were basically no differences in background between figures showing effect of miR-34a and control oligos. Consequently, mean background intensity was measured in experimental and control group and subtracted from these values.

Questionable experimental design:

The authors measure mRNA levels at different times (24 and 48h). Why such timelines where selected?

- above mentioned time points were selected as they are the most frequently used timepoints in this type of experiment and we wanted to make our date easily comparable with data of others researches focused on the similar topic.

Gao J, Li N, Dong Y, Li S, Xu L, Li X, Li Y, Li Z, Ng SS, Sung JJ, Shen L, Yu J. miR-34a-5p suppresses colorectal cancer metastasis and predicts recurrence in patients with stage II/III colorectal cancer. Oncogene. 2015 Jul 30;34(31):4142-52. doi: 10.1038/onc.2014.348.

https://pubmed.ncbi.nlm.nih.gov/25362853/

Peng Y, Fan JY, Xiong J, Lou Y, Zhu Y. miR-34a Enhances the Susceptibility of Gastric Cancer to Platycodin D by Targeting Survivin. Pathobiology. 2019;86(5-6):296-305. doi: 10.1159/000502913. 

https://pubmed.ncbi.nlm.nih.gov/31711057/

Wang C, Xin H, Yan G, Liu Z. NONHSAG028908.3 sponges miR‑34a‑5p to promote growth of colorectal cancer via targeting ALDOA. Oncol Rep. 2023 May;49(5):89. doi: 10.3892/or.2023.8526.

https://pubmed.ncbi.nlm.nih.gov/36929422/

Zhang D, Zhou J, Dong M. Dysregulation of microRNA-34a expression in colorectal cancer inhibits the phosphorylation of FAK via VEGF. Dig Dis Sci. 2014 May;59(5):958-67. doi: 10.1007/s10620-013-2983-4.

https://pubmed.ncbi.nlm.nih.gov/24370784/

Zhang X, Ai F, Li X, Tian L, Wang X, Shen S, Liu F. MicroRNA-34a suppresses colorectal cancer metastasis by regulating Notch signaling. Oncol Lett. 2017 Aug;14(2):2325-2333. doi: 10.3892/ol.2017.6444.

https://pubmed.ncbi.nlm.nih.gov/28781671/

The present study lack of any conclusions!!!!

- thank you very much for this comment, the section “Conclusions” was added after the section „Discussion“.

---

## [Decision Letter · Decision Letter 1]

5 Sep 2023

PONE-D-23-17825R1Effect of miR-34a on expression of clock and clock-controlled genes in human cancer DLD1 and Lovo cells with different backgrounds in respect to p53 functionality and 17β-estradiol-mediated regulationPLOS ONE

Dear Dr. Herichova,

Thank you for submitting your manuscript to PLOS ONE. After careful consideration, we feel that it has merit but does not fully meet PLOS ONE’s publication criteria as it currently stands. Therefore, we invite you to submit a revised version of the manuscript that addresses the points raised during the review process.

We look forward to receiving your revised manuscript.

Kind regards,

Shiki Fujino, M.D.

Academic Editor

PLOS ONE

Journal Requirements:

Reviewers' comments:

Reviewer's Responses to Questions

**Comments to the Author**

1. If the authors have adequately addressed your comments raised in a previous round of review and you feel that this manuscript is now acceptable for publication, you may indicate that here to bypass the “Comments to the Author” section, enter your conflict of interest statement in the “Confidential to Editor” section, and submit your "Accept" recommendation.

Reviewer #1: (No Response)

Reviewer #2: (No Response)

Reviewer #3: All comments have been addressed

2. Is the manuscript technically sound, and do the data support the conclusions?

Reviewer #1: No

Reviewer #2: Yes

Reviewer #3: Yes

3. Has the statistical analysis been performed appropriately and rigorously? 

Reviewer #1: I Don't Know

Reviewer #2: Yes

Reviewer #3: Yes

4. Have the authors made all data underlying the findings in their manuscript fully available?

Reviewer #1: Yes

Reviewer #2: Yes

Reviewer #3: Yes

5. Is the manuscript presented in an intelligible fashion and written in standard English?

Reviewer #1: No

Reviewer #2: Yes

Reviewer #3: Yes

6. Review Comments to the Author

Reviewer #1: Dear authors,

there are still many correction to do, specially in the discussion.

Line 373: when you refer to another work, you should write "has been" instead od "was".

Line 380;potentiate THE effects of each other.

Line 384: As under THE present experimental conditions miR-34a induced a decrease in per2 expression in DLD1 and....... I canceled "the"

line 403: The following sentence is not clear.

If you did the analysis in only DLD1 cells, how can you state that this effect is independent of the p53 status? "In our study administration of miR-34a significantly inhibited cell proliferation measured by an MTS assay without regard to the mutated form of p53 in DLD1 cells."

Line 413: It was shown that silencing of sirt1.......if this stament is refered in ref 28, you should say" It HAS been shown......

413 upregulates and overexpression of sirt1 causes a decrease in per2 expression.Line 418:r"recent experimental condition " change recent for i"n the present experimental conditions"

Line 433:"It seems that administration of ectopic miR-34a does not induce wound closure directly but via the MDM4/p53 pathway" why?

Line 435: You should not begin a paraghraph with "Morover" Just continue in the previous line.

Line 446: change In recent study miR-34a inhibited expression" for " In the present study"

Line 455: "A recent study supports" Do you mean " In the present study"

Line 463: In a recent study " change to " In the present study"

Line 464 It should be: "In spite of that E2-induced cyclin D1,..."

Line 467:We suppose that induction of cyclin D1 expression was mediated by SIGNALING OF MINORIT esr1 receptors present in DLD1 cells [13,107].

Line 489:Among major limitations of THE PRESENT study Line 509:seems to BE independent of miR-34a

Apart from grammatical errors there are some things that are not very clear.

In the first paragraph you state that the expression of bmal1 and rev-erba are influenced by miR-34a independent of p53 status but the first is just affected in LoVo cells and the second one, just in DLD1 cells; this should be commented.

Also cyclin D1 is different in LDL1 and LoVo cells

That E2 did not influence expression of miR-34a and miR-34a administration did not influence expression of esr2 in DLD1 cells, does not mean that they cannot have a sinergetic effect upon other target genes investigated. From your results, you can only say that there is no sinergic effects of miR-34a and E2 in proliferation and migration of LDL1 cells.

Material and Methods

How many independent experiments did you performed for each analysis to get the results? This should be clarified.

Can you please clarify how did you determine the concentation of miR-34a? In my experience, by nanodrop, you can just see total RNA and suppose that if RNA is of good quality, also miRs will be OK.Which quantity of total RNA you used for the experiments.

Reviewer #2: Dear Authors,

You are provided adequate answers on the comments. I think you still need to stress that research data were obtained in male cell lines (at least in Conclusions Section).

Reviewer #3: The authors have significantly improved the present manuscript according my recommendations. In addition, some questions were addressed vigorously based on literature review.

7. PLOS authors have the option to publish the peer review history of their article (what does this mean?). If published, this will include your full peer review and any attached files.

Reviewer #1: No

Reviewer #2: No

Reviewer #3: No

---

## [Author Response · Author response to Decision Letter 1]

26 Sep 2023

Comments from the editorial board

 - We did our best to avoid retracted articles. We are not aware of any retracted article in our list of references. All cited articles were checked in the database http://retractiondatabase.org/RetractionSearch.aspx

Comments and Suggestions for Authors

Reviewer #1:

Dear reviewer,

Thank you very much for your valuable time, suggestions, constructive comments, advice and questions; we appreciate them very much. All your suggestions and comments were implemented into the text, and questions were answered; please see the detailed responses bellow. We hope that you will find the revised MS acceptable for publication. 

With many thanks and best regards,

The authors

Reviewer #1: Dear authors,

there are still many correction to do, specially in the discussion.

Line 373: when you refer to another work, you should write "has been" instead od "was".

Line 380;potentiate THE effects of each other.

Line 384: As under THE present experimental conditions miR-34a induced a decrease in per2 expression in DLD1 and....... I canceled "the"

line 403: The following sentence is not clear.

If you did the analysis in only DLD1 cells, how can you state that this effect is independent of the p53 status? "In our study administration of miR-34a significantly inhibited cell proliferation measured by an MTS assay without regard to the mutated form of p53 in DLD1 cells."

Line 413: It was shown that silencing of sirt1.......if this stament is refered in ref 28, you should say" It HAS been shown......

413 upregulates and overexpression of sirt1 causes a decrease in per2 expression.Line 418:r"recent experimental condition " change recent for i"n the present experimental conditions"

Line 433:"It seems that administration of ectopic miR-34a does not induce wound closure directly but via the MDM4/p53 pathway" why?

Line 435: You should not begin a paraghraph with "Morover" Just continue in the previous line.

Line 446: change In recent study miR-34a inhibited expression" for " In the present study"

Line 455: "A recent study supports" Do you mean " In the present study"

Line 463: In a recent study " change to " In the present study"

Line 464 It should be: "In spite of that E2-induced cyclin D1,..."

Line 467:We suppose that induction of cyclin D1 expression was mediated by SIGNALING OF MINORIT esr1 receptors present in DLD1 cells [13,107].

Line 489:Among major limitations of THE PRESENT study Line 509:seems to BE independent of miR-34a

Apart from grammatical errors there are some things that are not very clear.

- Thank you very much for the comments. All suggestions were accepted, and the text was corrected accordingly. The sentence ‘It seems that administration of ectopic miR-34a does not induce wound closure directly but via the MDM4/p53 pathway’ was omitted from the text. 

As we are not native English speakers, there will always be some imperfections in the text if write it without professional help. For this reason, we always order the services of a professional agency. Several departments of our university are more than satisfied with the editing performed by Proof-Reading-Service.com (PRS); please see the link below.

https://www.proof-reading-service.com/about-us/

This is company is located in England close to the city of Cambridge, and it ranked among the top 10 Online Proofreading Services of the year 2023.

https://www.papertrue.com/blog/the-top-10-online-proofreading-services/

The MS was edited by PRS before submission, and we have had it proofread a second time for grammar and sentence structure. We hope that the quality of the text is now sufficient for publication in the prominent and reputable journal PLOS ONE. Please see also the newly released certificate of editing.

In the first paragraph you state that the expression of bmal1 and rev-erba are influenced by miR-34a independent of p53 status but the first is just affected in LoVo cells and the second one, just in DLD1 cells; this should be commented.

Also cyclin D1 is different in LDL1 and LoVo cells

- Thank you for the comment. In the first paragraph of the discussion, we intended to deliver the major idea of the article, that the circadian oscillator can be influenced by miR-34a-5p without respect to p53 functionality. We do not claim that p53 cannot influence the circadian oscillator at all if it is in its unmutated form. However, if p53 is not working, miR-34a-5p can still modify clock gene expression. In the revised version, we went into more detail and added information about particular genes whose expression was influenced in both cell lines. We also specified the cell lines with respect to particular genes.

The statement that miR-34a can influence circadian oscillator functioning also issued from other papers and conference contributions of our group:

Olejárová S, Moravčík R, Herichová I. 2.4 GHz Electromagnetic Field Influences the Response of the Circadian Oscillator in the Colorectal Cancer Cell Line DLD1 to miR-34a-Mediated Regulation. Int J Mol Sci. 2022; 23: 13210. doi:10.3390/ijms232113210

Herichová I, Reis R, Hasáková K, Moravčík R, Vicián M, Zeman M: Sex dependent expression of clock gene in human colorectal cancer tissue. In: Proceeding of 96. Physiological days, Jessenius Faculty of Medicine in Martin UK, Martin, s. 27, 2020.

Jendrisková S, Herichová I: Effect of electromagnetic field and miR-34a-5p on migration and gene expression of colorectal cancer cell line DLD1. In: : Proceeding of 97. Physiological days, Institute of Physiology of the 1. faculty of medicine CU, Charles University, Praha, s. 42, 2022.

That E2 did not influence expression of miR-34a and miR-34a administration did not influence expression of esr2 in DLD1 cells, does not mean that they cannot have a sinergetic effect upon other target genes investigated. From your results, you can only say that there is no sinergic effects of miR-34a and E2 in proliferation and migration of LDL1 cells.

- Thank you for the comment. The sentence ‘E2 co-administration did not modulate the effects of miR-34a, therefore, for now we cannot support our hypothesis that differences observed...’

was changed to:

‘E2 co-administration did not modulate the effect of miR-34a on DLD1 proliferation, so for now we cannot support our hypothesis that differences observed... ’

Our statement that miR-34a and E2 do not act synergically is also based on the observation that while miR-34a inhibited the expression of cyclin D1, administration of E2 had the opposite effect.

Material and Methods

How many independent experiments did you performed for each analysis to get the results? This should be clarified.

- The sentence ‘Experiments were performed three times.’ was included in the section ‘Material and methods’.

Can you please clarify how did you determine the concentation of miR-34a? In my experience, by nanodrop, you can just see total RNA and suppose that if RNA is of good quality, also miRs will be OK.Which quantity of total RNA you used for the experiments.

- Thank you for the comment. We are sorry that the sentence ‘Micro RNA from the DLD1 cell line was isolated using RNAzol RT (Molecular Research Center). The miRNA concentration and purity were determined using a Simply Nano spectrometer (Ge Healthcare, USA).’ was unclear and probably led to the impression that the concentration of miR-34a was measured by nanodrop. As stated in the ‘Materials and methods’ section, we used RNAzol RT (Molecular Research Center) to isolate RNA. In particular, we used the ‘Protocol for isolation of large RNA and small RNA fraction’ that allows the separation of long and short RNA molecules; please see the link below.

https://www.mrcgene.com/wp-content/uploads/2017/04/RNAzolRTMarch2017.pdf

According to this protocol, it is recommended to measure the concentration of miRNAs similarly to the concentration of long RNA molecules, only that the 260/280 ratio is different. 

To avoid misunderstanding concerning miR-34a measurement, we used plural when describing miRNA isolation and made some other minor changes. We also included information about the particular protocol that was used for miRNA RT.

We did not use total RNA in reverse transcription. To synthesize cDNA from mRNA, we used 0.3 ug of the RNA fraction with long molecules, which is well within the range (1 pg–1 μg) of the kit mentioned in the section ‘Materials and methods’; please see link bellow.

https://worldwide.promega.com/-/media/files/resources/protocols/technical-manuals/0/improm-ii-reverse-transcription-system-protocol.pdf?rev=5d3ab0d332a543b9847784041cdcf152&sc_lang=en

The concentration of miR-34a was measured by real-time PCR using relative quantification. After miRNA isolation, sequences were extended by adenylation and adding a tag during reverse transcription. This was followed by a PCR reaction with two primers that allowed specific measurement of miR-34a-5p from pool of all isolated and transcribed miRNAs. The majority of the miRNA sequence is used for priming by the forward primer. The specificity of the reverse primer is based on partial overlap with the miRNA sequence. This technique is described in the paper (Barcells et al., 2011; 197 references in WOS) and has been adapted for commercial use by Qiagen (please see link below).

https://www.qiagen.com/us/products/discovery-and-translational-research/pcr-qpcr-dpcr/qpcr-assays-and-instruments/mirna-qpcr-assay-and-panels/mircury-lna-mirna-pcr-assays

The sequences of all primers are provided in supplementary table 1. This method of miR-34a-5p measurement has been used previously in the paper Hasakova et al. (2019). These details are now provided in the section ‘Materials and methods’.

Balcells I, Cirera S, Busk PK. Specific and sensitive quantitative RT-PCR of miRNAs with DNA primers. BMC Biotechnol. 2011; 11:70. doi: 10.1186/1472-6750-11-70.

Hasakova K, Reis R, Vician M, Zeman M, Herichova I. Expression of miR-34a-5p is up-regulated in human colorectal cancer and correlates with survival and clock gene PER2 expression. PLoS One. 2019; 14(10):e0224396. doi: 10.1371/journal.pone.0224396.

---

## [Editor Report · Decision Letter 2]

2 Oct 2023

Effect of miR-34a on the expression of clock and clock-controlled genes in DLD1 and Lovo human cancer cells with different backgrounds with respect to p53 functionality and 17β-estradiol-mediated regulation

PONE-D-23-17825R2

Dear Dr. Herichova,

We’re pleased to inform you that your manuscript has been judged scientifically suitable for publication and will be formally accepted for publication once it meets all outstanding technical requirements.

Kind regards,

Shiki Fujino, M.D.

Academic Editor

PLOS ONE

---

## [Editor Report · Acceptance letter]

6 Oct 2023

PONE-D-23-17825R2 

Effect of miR-34a on the expression of clock and clock-controlled genes in DLD1 and Lovo human cancer cells with different backgrounds with respect to p53 functionality
and 17β-estradiol-mediated regulation 

Dear Dr. Herichova:

I'm pleased to inform you that your manuscript has been deemed suitable for publication in PLOS ONE. Congratulations! Your manuscript is now with our production department. 

Kind regards, 

on behalf of

Dr. Shiki Fujino 

Academic Editor

PLOS ONE